



# Detectability of $CO_2$ emission plumes of cities and power plants with the Copernicus Anthropogenic $CO_2$ Monitoring (CO2M) mission

Gerrit Kuhlmann[1], Grégoire Broquet[2], Julia Marshall[3], Valentin Clément[4,5], Armin Löscher[6], Yasjka Meijer[6], and Dominik Brunner[1]

[1]Empa, Swiss Federal Laboratories for Materials Science and Technology, Dübendorf, Switzerland
[2]Laboratoire des Sciences du Climat et de l'Environment, LSCE/IPSL, CEA-CNRS-UVSQ, Université Paris-Saclay, Gif-sur-Yvette, France
[3]Max Planck Institute for Biogeochemistry (MPI-BGC), Jena, Germany
[4]Center for Climate Systems Modelling (C2SM), ETH Zurich, Zurich, Switzerland
[5]MeteoSwiss, Kloten, Switzerland
[6]European Space Agency (ESA), ESTEC, Noordwijk, The Netherlands

**Correspondence:** Gerrit Kuhlmann (gerrit.kuhlmann@empa.ch)

**Abstract.** High-resolution atmospheric transport simulations were used to investigate the potential for detecting carbon dioxide ($CO_2$) plumes of the city of Berlin and neighboring power stations with the Copernicus Anthropogenic Carbon Dioxide Monitoring (CO2M) mission, which is a proposed constellation of $CO_2$ satellites with imaging capabilities. The potential for detecting plumes was studied for satellite images of $CO_2$ alone or in combination with images of nitrogen dioxide ($NO_2$) and

carbon monoxide (CO) to investigate the added value of measurements of other gases co-emitted with $CO_2$ that have better signal-to-noise ratios. The additional $NO_2$ and CO images were either generated for instruments on the same CO2M satellites ($2\times2$ km² resolution) or for the Sentinel-5 instrument ($7\times7$ km²) assumed to fly two hours earlier than CO2M. Realistic $CO_2$, CO and $NO_2$ fields were simulated at $1\times1$ km² horizontal resolution with COSMO-GHG model for the year 2015, and used as input for an orbit simulator to generate synthetic observations of columns of $CO_2$, CO and $NO_2$ for constellations of up

to six satellites. A new plume detection algorithm was applied to detect coherent structures in the images of $CO_2$, $NO_2$ or CO against instrument noise and variability in background levels. Although six satellites with an assumed swath of 250 km were sufficient to overpass Berlin on a daily basis, only about 50 out of 365 plumes per year could be observed in conditions suitable for emission estimation due to frequent cloud cover. With the $CO_2$ instrument only 6 and 16 of these 50 plumes could be detected assuming a high ($\sigma_{VEG50}$ = 1.0 ppm) and low noise ($\sigma_{VEG50}$ = 0.5 ppm) scenario, respectively, because the $CO_2$

signals were often too weak. A CO instrument with specifications similar to the Sentinel-5 mission performed worse than the $CO_2$ instrument, while the number of detectable plumes could be significantly increased to about 35 plumes with an $NO_2$ instrument. Using $NO_2$ observations from the Sentinel-5 platform instead resulted in a significant spatial mismatch between $NO_2$ and $CO_2$ plumes due to the two hours time difference between Sentinel-5 and CO2M. The plumes of the coal-fired power plant Jänschwalde were easier to detect with the $CO_2$ instrument (about 40-45 plumes per year), but again, an $NO_2$ instrument

performed significantly better (about 70 plumes). Auxiliary measurements of $NO_2$ were thus found to greatly enhance the capability of detecting the location of $CO_2$ plumes, which will be invaluable for the quantification of $CO_2$ emissions from large point sources.



# 1 Introduction

The signatory countries of the Paris climate agreement have set ambitious goals to reduce $CO_2$ emissions and limit global warming to below $2°$ C above pre-industrial levels (UNFCCC, 2015). The efficient implementation and management of long-term policies will require consistent, reliable, and timely information on $CO_2$ emissions (Ciais et al., 2015; Pinty et al., 2018).

The majority of these emissions are concentrated on a small fraction of the globe, primarily on cities and power plants. Acknowledging their important role, cities have started to devise policies for cutting $CO_2$ emissions often surpassing the reduction targets of the respective countries (e.g. C40 cities, 2018). However, cities are currently lacking an independent $CO_2$ emission monitoring system to evaluate their policies.

The European Space Agency (ESA) and the European Commission (EC) therefore propose the Copernicus Anthropogenic
Carbon Dioxide Monitoring (CO2M) mission, a constellation of $CO_2$ satellites with imaging capability, to support the quantification of anthropogenic $CO_2$ fluxes and to assist greenhouse gas mitigation policies at the national, city and facility level. The satellites are envisioned as an essential component of a $CO_2$ observation system to be established under Europe's Earth observation program Copernicus (Ciais et al., 2015; Pinty et al., 2018). The system would allow for observing $CO_2$ plumes of individual point sources such as large cities and power plants and for quantifying the respective emissions during single satel-
lite overpasses (Bovensmann et al., 2010; Pillai et al., 2016; Velazco et al., 2011). While the potential for detecting strong $CO_2$ plumes of megacities and large point sources has been demonstrated for the Orbiting Carbon Observatory 2 (OCO-2, Crisp et al. (2017)) (Nassar et al., 2017; Reuter et al., 2019), a major challenge is to accurately determine the location of weaker $CO_2$ plumes with signal-to-noise ratios near or below the detection limit for single pixels. The detection of such weak plumes is additionally challenged by the interference with signals from biospheric $CO_2$ fluxes and other anthropogenic sources in the
vicinity of the target. Therefore, measurements of auxiliary trace gases co-emitted with $CO_2$ but little affected by biospheric processes such as carbon monoxide (CO) and nitrogen dioxide ($NO_2$) were proposed to help separate anthropogenic from biospheric $CO_2$ signals (Ciais et al., 2015).

This study presents results from the SMARTCARB project (use of Satellite Measurements of Auxiliary Reactive Trace gases for fossil fuel CARBon dioxide emission estimation (Kuhlmann et al., 2019)), which aimed to assess the potential synergies
of measurements of CO and $NO_2$ for observing and quantifying $CO_2$ emissions and to help define the required satellite specifications. To address these questions, Observing System Simulation Experiments (OSSE) were conducted, for which synthetic satellite observations were generated from high-resolution atmospheric transport simulations. The model domain was centered on the city of Berlin and also covered several nearby power plants. Similar simulations were already performed in previous OSSEs (Pillai et al., 2016; Broquet et al., 2018), but they did not have a comparable spatial resolution or temporal
extent or did not cover the additional species $NO_2$ and CO as investigated here.

In a companion paper Brunner et al. (2019) presented the overall model setup and emphasized the importance of properly accounting for the vertical placement of $CO_2$ emissions from large point sources in atmospheric $CO_2$ simulations. Here, we investigate whether and how often the $CO_2$ (or $NO_2$ or CO) plume of a city or a power plant can be detected during a year depending on the size of the satellite constellation and on instrument specifications. For this purpose, a novel plume detection



algorithm was developed which identifies the plume signals against instrument noise and background variability. In a follow-up study, we will quantify the emissions from Berlin and a few power plants in the model domain from the synthetic satellite observations using both inverse and mass-balance approaches, building on the plume detection presented here.

## 2 Data and methods

In a satellite image, a plume may be defined as a collection of spatially connected pixels with elevated signals starting at a source. Whether and how frequently the plume of a given source can be detected depends on several, partly interdependent factors:

- The number of satellites and the instrument's swath width, as they determine the number of overpasses over the plume and how much of the plume is visible in the satellite image.

- The intensity of the emission source, which affects the amplitude of the enhancement above background.

- The meteorological conditions, notably wind speed and turbulence, which determine the dilution and dispersion of the emissions.

- The single sounding precision of the instrument, which determines if the enhancement within the plume can be detected.

- The variability of the background, which is caused by anthropogenic emissions and biospheric fluxes upstream or downstream of the considered source and which is additionally affected by meteorology.

- The presence of clouds partially or fully obscuring the plume.

Since most of these factors vary with season, the detectability also depends on the time of the year. Therefore, long simulations covering a full year were conducted.

Because the detection of weak anthropogenic $CO_2$ plumes is affected by interference with biospheric $CO_2$ signals, auxiliary trace gases co-emitted with $CO_2$ could be used for locating the $CO_2$ plume in the satellite image. However, this requires that the plumes of $CO_2$ and of the auxiliary trace gas are spatially congruent. This is the case when they are emitted from the same source, for example, a power plant. The situation is more complex for cities where the emissions originate from different sectors (industry, heating, transport etc.) that emit at different altitude levels and have different emission ratios of $CO_2$ to $NO_2$ or CO (Brunner et al., 2019). In this study, we therefore carefully consider the vertical distribution of emissions for different species, which makes it possible to test for congruence.

### 2.1 Synthetic satellite observations

#### 2.1.1 Model simulations

The synthetic satellite observations were generated from high-resolution simulations conducted with the COSMO-GHG model. COSMO is a hydrostatic, limit-area model developed by the Consortium for Small-scale Modeling (Baldauf et al., 2011), for

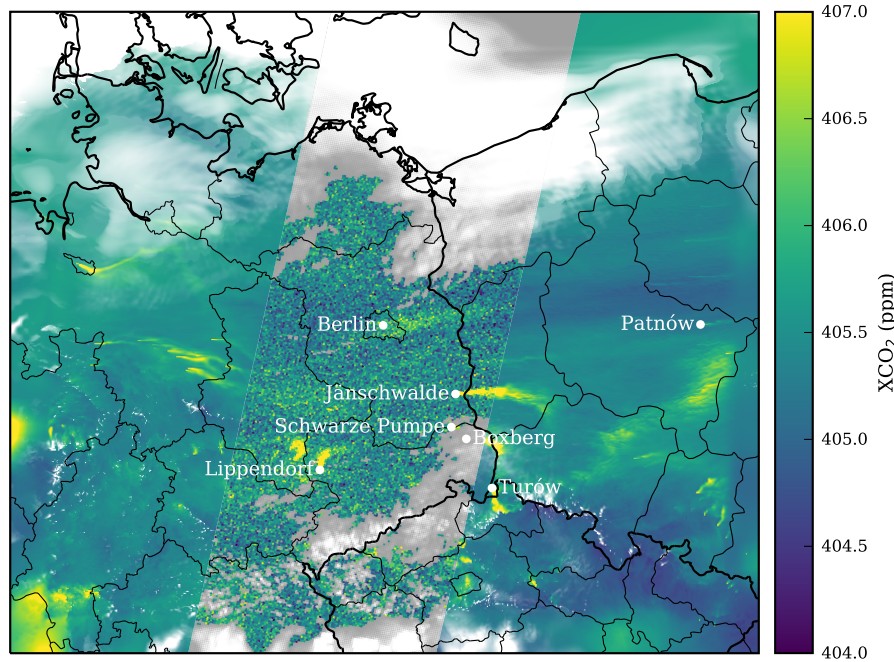

**Figure 1.** Simulated $XCO_2$ field on 23 April 2015 in the SMARTCARB model domain overlaid with an example of a 250 km wide swath of the planned Sentinel $CO_2$ instrument (low noise scenario). Missing $CO_2$ measurements are shown in gray. Cloud cover is overlayed in white with transparency corresponding to total cloud fraction.

which an extension has been developed for the simulation of greenhouse gases (COSMO-GHG)(Oney et al., 2015; Liu et al., 2017).

COSMO-GHG was set up to simulate $CO_2$, CO and $NO_x$ concentration fields for nearly the complete year 2015 (1 January - 25 December). The model domain extended about 750 km in east-west and 650 km in south-north direction. It was centered

over the city of Berlin and also covered numerous power plants in Germany and neighboring countries. The spatial resolution was 1.1 km × 1.1 km horizontally with 60 vertical levels up to an altitude of 24 km. Figure 1 presents the model domain and marks the location of Berlin and the six largest coal-fired power plants. The detailed model setup is described by Brunner et al. (2019).

Initial and lateral boundary conditions (ICBC) for meteorological variables were provided by the operational European

COSMO-7 analyses of MeteoSwiss with hourly temporal and 7 km horizontal resolution. For the tracers, ICBCs were obtained from the European Center for Medium Range Weather Forecast (ECMWF) through the European Earth observation program Copernicus. $CO_2$ and CO boundary conditions were taken from a global free-running $CO_2$ simulation with 137 levels and about 15 km horizontal resolution (T1279 spectral resolution, experiment "gf39", class "rd")(Agustí-Panareda et al., 2014). For NO and $NO_2$, boundary conditions were taken from ECMWF's operational global forecasts for aerosol and chemical

species with 60 vertical levels and a horizontal resolution of about 60 km (T255 resolution, experiment "0001", class "mc") (Flemming et al., 2015).





**Table 1.** Emissions of largest power plants in model domain according to TNO/MACC-3 inventory for the year 2011 as used in this study.

| Power plant | $CO_2$ (Mt yr$^{-1}$) | $NO_x$ (kt yr$^{-1}$) | CO (kt yr$^{-1}$) |
|---|---|---|---|
| Jänschwalde | 33.3 | 26.9 | 44.1 |
| Boxberg | 19.0 | 15.4 | 14.4 |
| Lippendorf | 15.3 | 12.3 | 2.7 |
| Turów | 8.7 | 13.1 | 1.3 |
| Schwarze Pumpe | 8.2 | 6.6 | 5.3 |

Anthropogenic emissions were obtained by combining the Netherlands Organisation for Applied Scientific Research European Monitoring Atmospheric Composition and Climate version 3 (TNO/MACC-3) inventory (Kuenen et al., 2014, for Version 2) with a detailed inventory provided by the city of Berlin. The temporal variability of emissions was accounted for by applying diurnal, weekly, and seasonal cycles according to Selected Nomenclature for Air Pollution (SNAP) categories. Furthermore,
emissions were vertically distributed using specific vertical profiles for the different emissions categories and plume rise calculations for six largest power plants and the major point sources in Berlin (Brunner et al., 2019). Hourly biospheric fluxes both photosynthesis and respiration were generated with the vegetation photosynthesis and respiration model (VPRM) at the resolution of the COSMO model (Mahadevan et al., 2008).

According to the official inventory of the city of Berlin, total annual $CO_2$ emissions of Berlin were $16.9 \, \text{Mt} \, CO_2 \, \text{yr}^{-1}$ (in
the reference year 2012 of the inventory). This is about a factor two smaller than in previous studies, e.g. in the LOGOFLUX project (Chimot et al., 2013; Bacour et al., 2015; Pillai et al., 2016), which relied on unrealistically high emissions as provided by the global EDGAR inventory (Version 4.1). Due to the diurnal cycle of emissions, emissions were somewhat larger (about $20.0 \, \text{Mt} \, CO_2 \, \text{yr}^{-1}$) around the time of the satellite overpasses (10-11 UTC). Table 1 summarizes the $CO_2$, $NO_x$ and CO emissions of the five largest power plants in the domain.

The simulations included a total of 50 different passively transported tracers representing the three different gases further divided into different sources, release times or release altitudes. This also included background tracers constrained at the lateral boundaries by the global-scale models and two tracers for biospheric respiration and photosynthesis for $CO_2$. Due to the reactivity of $NO_x$, five different $NO_x$ tracers with e-folding lifetimes of 2, 4, 12 and 24 hours and infinity were included, considering that the lifetime of $NO_x$ varies between about 2 and 24 hours (Schaub et al., 2007). The full list of tracers is
provided in the SMARTCARB final report (Kuhlmann et al., 2019, p. 15f)

In this study we use the following seven tracers that have been computed from the 50 tracers in the simulations:

– X_BER: concentrations from time-varying emissions of Berlin,

– X_PP: concentrations from time-varying emissions from the six largest power plants in the model domain,

– X_ANTH: concentrations from other anthropogenic sources in the domain excluding emissions of Berlin (X_BER) and
the six largest power plants (X_PP),





**Table 2.** Satellite platforms and orbits.

| Parameter | CO2M satellite | MetOp-SG-A |
|---|---|---|
| Orbit type | Sun-synchronous | Sun-synchronous |
| Inclination | 97.77° | 98.7° |
| Orbits per day | 14+10/11 | 14+6/29 |
| Cycle duration | 11 days | 29 days |
| Cycle length | 164 orbits | 412 orbits |
| Altitude | 602.24 km | 830.16 km |
| Orbit Period | 96.58 minutes | 101.36 minutes |
| Local time in descending node (equator crossing time) | 11:30 hrs | 9:30 hrs |

**Table 3.** Observation geometries for instruments on the CO2M satellite and Sentinel-5.

| Parameter | CO2M satellite | Sentinel-5 |
|---|---|---|
| Number of across-track pixels | 125 | 208 |
| Swath | 250 km | 2670 km |
| Field of view | 23.22° | 107.1° |
| Pixel size | $2 \times 2$ km$^2$ | from $7.5 \times 7.5$ km$^2$ (nadir) to $35\times7.5$ km$^2$ (swath edge) |
| Along-track sampling time | 0.286 seconds | 1.13 seconds |

– X_BIO: concentrations from local biospheric fluxes, i.e. respiration and photosynthesis within the domain (only for $CO_2$),

– X_TOT: concentrations from all emissions and biospheric fluxes as well as inflow from lateral boundaries,

– X_BER_BG: concentrations from emissions, fluxes and lateral boundaries excluding emissions from Berlin (= X_TOT – X_BER),

– X_PP_BG: concentrations from emissions, fluxes and lateral boundaries excluding emissions from the six major power plants (= X_TOT – X_PP),

where X is $CO_2$, CO or $NO_2$. For $NO_2$ only the tracers with a lifetime of 4 hours were used. Note that only the sum of the emissions from the six power plants was simulated but not the power plants individually, which often complicated the analysis due to overlapping plumes. For the analysis, the three-dimensional model fields were vertically integrated to compute column-averaged dry air mole fractions of $CO_2$ ($XCO_2$). Likewise, tropospheric CO and $NO_2$ vertical column densities (VCD) were generated by considering only the model fields below 10 km altitude.

### 2.1.2 Satellite instrument scenarios

For the $CO_2$, CO and $NO_2$ satellite observations, different instrument scenarios were prescribed by ESA for the study in terms of orbit, spatial resolution and spatial and temporal coverage. For the scenarios, two representative platforms were considered:





(a) (b)

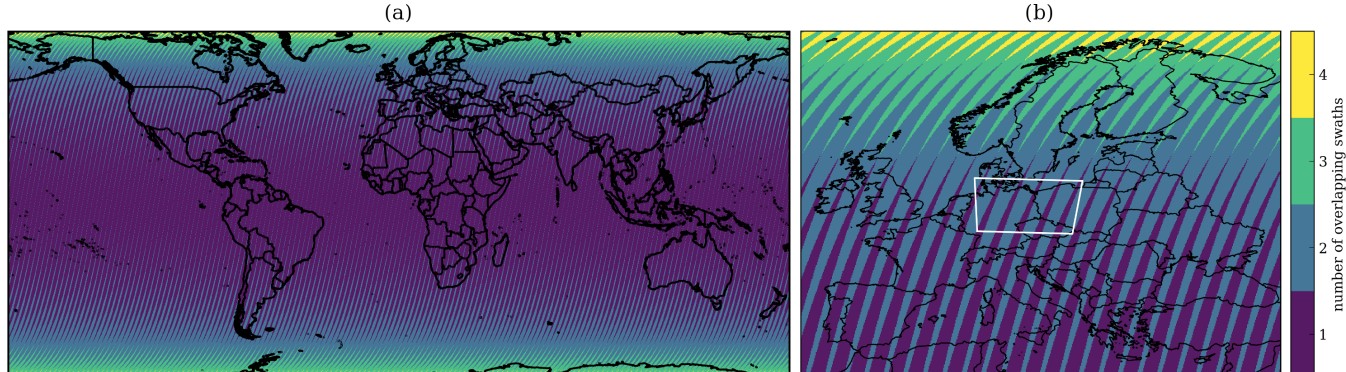

**Figure 2.** Spatial coverage of one CO2M satellite within its 11-day repeat cycle (a) globally and (b) over Europe. The white square marks the COSMO-GHG model domain in which the number of overpasses is either one or two. The exact locations of the "stripes" are arbitrary and depend on the equator starting longitude (here: $0\,^{\circ}$E) of the satellite.

MetOp-SG-A (Meteorological Operational Satellite – Second Generation - A) and the proposed CO2M satellite. MetOp-SG-A is a series of meteorological satellites in low earth orbit that will carry the Sentinel-5 instrument measuring, among others, $NO_2$ and CO VCDs. The CO2M mission is a proposed platform with a $CO_2$ instrument as the main payload and optionally with instruments for $NO_2$, CO and aerosols. Both satellites will be in sun-synchronous orbits but with different equator crossing

times and repeat cycles.

In addition to a single satellite, the potential of a constellation of multiple CO2M satellites was also studied. The basic assumption for a constellation is that the individual satellites would be spaced equally in a common (circular) orbit, for instance separated by 120° in case of 3 satellites. Here, we analyze constellations between one and six satellites. The individual satellites can be distinguished by their starting longitude at the equator of the first orbit in the repeat cycle.

For the computation of orbits, we adopted the orbit simulator of the Netherlands Institute for Space Research (SRON). Since this simulator makes a few simplifying assumptions such as circular orbits and tiled ground pixels, satellite and instrument parameters were slightly modified to preserve essential parameters. In particular, orbit periods were calculated to match a given cycle duration and length. The period then determines the altitude and inclination of a circular, sun-synchronous orbit. The altitude of the circular orbits is slightly larger than the typically used mean altitude for elliptic orbits. Since the altitude

affects the size of the ground pixels and the width of the swath, field of view and along-track sampling time were set to match exactly the prescribed pixel size at sub-satellite point as well as the prescribed swath width. As a result, the number of across-track pixels for Sentinel-5 did not match exactly the number of pixels for the real Sentinel-5 instrument.

Tables 2 and 3 summarize the orbits and viewing geometries of the two satellites. The CO2M satellite is assumed to have a 250-km wide swath and 11-day repeat cycle. Within the 11-day cycle, the instrument provides nearly global spatial coverage

(Fig. 2). For locations in the SMARTCARB model domain, either one or two overpasses occur during the 11-day repeat cycle depending on the equator starting longitude of the satellite. Sentinel-5 has a much wider swath of $2650\,\mathrm{km}$ resulting in near-daily global coverage (not shown).



**Table 4.** Instrument uncertainty scenarios. VEG50 refers to a reference scene with a surface albedo of a vegetated surface and a solar zenith angle (SZA) of $50°$.

| Scenario name | Species | Satellite(s) | Reference noise ($\sigma_{\mathrm{VEG50}}$ / $\sigma_{\mathrm{ref}}$) | |
| --- | --- | --- | --- | --- |
| | | | absolute[1] | relative[1] |
| $CO_2$ low noise | $CO_2$ | CO2M | 0.5 ppm | - |
| $CO_2$ medium noise | $CO_2$ | CO2M | 0.7 ppm | - |
| $CO_2$ high noise | $CO_2$ | CO2M | 1.0 ppm | - |
| $NO_2$ low noise | $NO_2$ | CO2M | $1.0 \times 10^{15}$ molec. cm$^{-2}$ | 15% |
| $NO_2$ high noise | $NO_2$ | CO2M | $2.0 \times 10^{15}$ molec. cm$^{-2}$ | 20% |
| $NO_2$ Sentinel-5 | $NO_2$ | Sentinel-5 | $1.3 \times 10^{15}$ molec. cm$^{-2}$ | 20% |
| CO low noise | CO | Sentinel-5 and CO2M | $4.0 \times 10^{17}$ molec. cm$^{-2}$ | 10% |
| CO high noise | CO | Sentinel-5 and CO2M | $4.0 \times 10^{17}$ molec. cm$^{-2}$ | 20% |

[1] whichever is larger

$XCO_2$, CO and $NO_2$ column densities were sampled along the satellite swath for one year using the tracers from the COSMO-GHG simulations. For CO2M, $XCO_2$, CO and $NO_2$ columns were mapped onto the $2\,km \times 2\,km$ size pixels along the 250-km wide swath.

For Sentinel-5, CO and $NO_2$ columns were sampled with up to $7.5\,km \times 7.5\,km$ resolution along the 2670-km wide swath

of the Sentinel-5 instrument. Due to the wide swath, the pixel sizes grow towards the edge of the swath. In this study, only the spatial overlap between Sentinel-5 and CO2M were of interest, because Sentinel-5 was used for detecting the $CO_2$ emission plumes inside the swath of CO2M.

### 2.1.3 Instrument error characteristics

The error characteristics of the $CO_2$, $NO_2$ and CO instruments were specified in collaboration with ESA based on previous

studies for Carbonsat (Buchwitz et al., 2013) and on performance requirements for Sentinel-5. For the instruments on the CO2M satellites, two or three scenarios were included in order to cover a realistic range between more or less demanding instruments. Table 4 summarizes the single sounding precision of the different instruments.

For $XCO_2$, three different uncertainty scenarios were considered which relate back to the performance estimates derived for the Carbonsat mission concept. A detailed error budget for Carbonsat was presented in the Carbonsat Report for Mission

Selection (ESA, 2015). In the LOGOFLUX study, error parameterization formulas (EPF) for random and systematic errors were developed, which account for errors introduced by solar zenith angle (SZA), surface reflectance in the near infrared (NIR) and shortwave infrared (SWIR-1), cirrus clouds, and aerosol optical depth (Buchwitz et al., 2013).

Here, the same EPFs were adopted but only applied to compute random errors. These were calculated based on SZA and surface reflectance in the NIR and SWIR-1 band. Surface reflectances were taken from the MODIS MCD43A3 product (Version

006) at 1 km spatial resolution (Schaaf and Wang, 2015). A detailed consideration of cirrus clouds and aerosols and their impact



on systematic errors was outside the scope of the study as it would have required the collection and processing of a large amount of additional data. The possible impact of not considering systematic errors will briefly be discussed in Sect. 4.

The random error calculated with the EPFs for the so-called vegetation-50 scenario (VEG50, i.e. vegetation albedo and SZA of 50°) is about 1.5 ppm. In the model domain, mean random errors are slightly smaller at 1.3 ppm. To obtain random errors for the three instrument scenarios with $\sigma_{\mathrm{VEG50}}$ of 0.5, 0.7 and 1.0 ppm, the computed errors were divided by 3.0, 2.14 and 1.5, respectively.

For $NO_2$ VCDs, the overall uncertainties are due to (a) measurement noise and spectral fitting affecting the slant column densities, (b) uncertainties related to the separation of the stratospheric and tropospheric column and (c) uncertainties in the auxiliary parameters used for air mass factor (AMF) calculations such as clouds, surface reflectance, a priori profile shapes and aerosols (Boersma et al., 2004). The total uncertainties are dominated by uncertainties from spectral fitting for background pixels and by uncertainties in AMF calculations for polluted pixels, respectively. Typical spectral fitting uncertainty of previous instruments such as OMI were of the order of $1\text{-}2\times10^{15}$ molecules $\mathrm{cm}^{-2}$ and AMF uncertainties of the order of 15-20%. These ranges were used to define two different scenarios for a possible CO2M $NO_2$ instrument (Table 4). For the Sentinel-5 UVNIS instrument, we assumed a relative uncertainty of 20% and a minimum uncertainty of $1.3\times10^{15}$ molecules per $\mathrm{cm}^{-2}$. In the presence of clouds, the reference noise was increased using the empirical formula developed by Wenig et al. (2008). For a cloud fraction of 30%, random noise is approximately doubled.

For CO VCDs, the total uncertainty depends on the (a) fitting noise and (b) a priori CO and $CH_4$ profiles and (c) surface reflectance, aerosols and clouds. We assumed a single sounding precision of $4.0\times10^{17}$ molecules per $\mathrm{cm}^2$ and a relative precision of 10% and 20% for both Sentinel-5 and the CO2M mission.

### 2.1.4 Cloud filtering

Satellite observations require filtering for clouds, which significantly reduces the number of observations available for plume detection. For the $CO_2$ product, we removed all $CO_2$ pixels with cloud fractions larger than 1%, because the $CO_2$ requires rigorous cloud filtering (Taylor et al., 2016). The $NO_2$ retrieval can tolerate larger errors and is therefore less sensitive to clouds. For the $NO_2$ product, we used a cloud threshold of 30% as often applied in satellite $NO_2$ studies (e.g. Boersma et al., 2011). For CO, a cloud threshold of 5% was used, which is motivated by the cloud threshold used for the MOPITT CO product (Deeter et al., 2017; MOPITT Algorithm Development Team, 2017). All cloud information in this study was obtained from COSMO-GHG, i.e. the same model as used for the tracer transport simulations.

## 2.2 Plume detection algorithm

### 2.2.1 Algorithm

We developed a new plume detection algorithm that uses a statistical test to detect signal enhancements which are significant with respect to instrument noise and variability in background levels. The plume is then identified as a coherent structure of significant pixels. The algorithm involves three processing steps as laid out in Fig. 3.

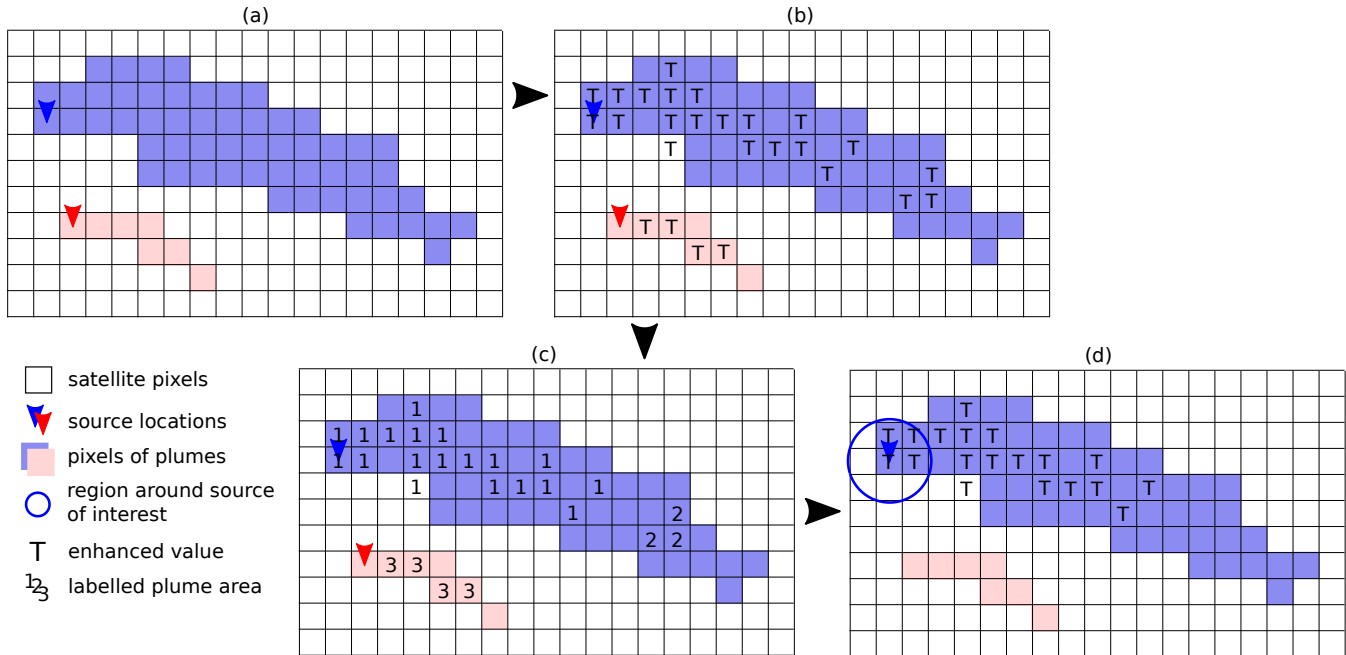

**Figure 3.** Schematic of the processing steps of the plume detection algorithm: (a) A large (red) and small plume (blue) with sources marked by arrows are located within the satellite overpass. (b) Pixels detected based on Z-test are marked with "T" for True. (c) Connected pixels are given a unique number each denoting a different plume. (d) All plumes not connected with the source of interest (blue circle) are rejected.

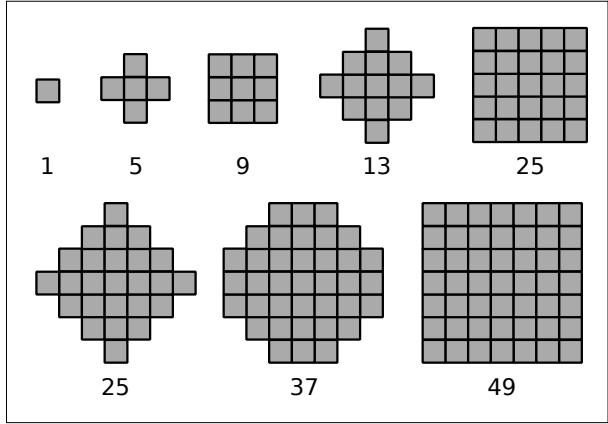

**Figure 4.** Examples of neighborhoods with different sizes ($n_s$) used for calculating the local mean.

A key feature of the algorithm is that the signal of a single pixel is replaced by a spatial average of pixels in a defined neighborhood. This allows identifying weak plumes with signals of individual pixels well below instrument noise, but also bares the risk of diluting the signal at the plume edges. Figure 4 shows examples of neighborhoods with different sizes $n_s$ that have been used for testing the algorithm. A large neighborhood results in a stronger smoothing and may therefore produce a



larger number of false positives, i.e. pixels outside of the plume that are wrongly assigned to the plume. An ideal neighborhood size balances the need for sensitive plume detection with the requirement of a low fraction of false positives.

Whether a signal enhancement is detectable is primarily determined by the signal-to-noise ratio (SNR) at a satellite pixel. Thereby, the signal is the enhancement above background due to the plume, and the noise is composed of both instrument

noise and spatial variability in the background. Since the background, i.e. the trace gas field in the absence of the plume, can not directly be observed, it needs to be estimated either from the observable trace gas field surrounding the plume or from a climatology or a model. Instrument noise and background variability can have both random and systematic components. The random component would reduce with the inverse square root of the number of pixels $n$ of the neighborhood, whereas the systematic error would be approximately independent of $n$.

The SNR is thus computed as

$$\text{SNR} = \frac{X_{\text{obs}} - X_{\text{bg}}}{\sqrt{\frac{\sigma_{\text{rand}}^2}{n} + \sigma_{\text{sys}}^2}} \qquad (1)$$

where $X_{\text{obs}}$ are the observed trace gas values, $X_{\text{bg}}$ is the estimated background value, and $\sigma_{\text{rand}}$ and $\sigma_{\text{sys}}$ are the random and systematic errors, respectively.

The first step of the the plume detection algorithm finds satellite pixels with $CO_2$, CO or $NO_2$ values significantly larger

than the background field by using a statistical Z-test, for which the distribution of the test statistics can be approximated by a normal distribution (e.g. v. Storch and Zwiers, 2003). The Z-test calculates the SNR for each satellite pixel and compares it to the z-value $z(q)$ for which $X_{\text{obs}}$ is larger than $X_{\text{bg}}$ with probability $q$:

$$\text{SNR} \geq z(q). \qquad (2)$$

To compute the SNRs from the satellite image, we computed observed values $X_{\text{obs}}$ as the local means of a neighborhood of

size $n_s$ (Fig. 4). For sake of simplicity, the background $X_{\text{bg}}$ in this study was estimated from the X_BER_BG and X_PP_BG model tracers for a $200 \times 200$ km$^2$ square centered on the city of Berlin and a $100 \times 100$km$^2$ square centered on each power plants. The implications of this assumption are discussed in Sect. 4.

For random and systematic errors we assumed that the instrument uncertainty is purely random and that the background variability is purely systematic. For the instrument uncertainty, random errors as listed in Table 4 were used, which reduce with

the number of valid pixels in the local neighborhood $n$ due to the inverse scaling by the number of pixels. The background uncertainty $\sigma_{bg}^2$ was computed from the spatial variance of the background model tracer in a fixed domain surrounding the source as described above. Since it is assumed to be systematic, it does not decrease with the size of the neighborhood.

The result of the Z-test is a binary image with "true" values where pixels are significantly enhanced above the background and "false" values where they aren't (Fig. 3b). Since the local mean can still be computed for missing center pixels - using

neighboring pixels - missing pixels can also be detected as enhanced above the background.





In the second step, pixels that are enhanced ("true") and connected are assumed to belong to the same plume. Regions of connected pixels are identified using a Moore neighborhood, where each pixel has eight potential neighbors, and are labelled with a unique integer value (Fig. 3c).

Finally, in the third step, all connected regions that do not intersect with the source region are removed leaving only regions that overlap with the source of the plume (Fig. 3d). For cities, the "source region" is defined by a circle with a radius of 15 km and for point sources by a circle with a radius of 5 km. The last step may remove regions that are part of the real plume but separated from the source by weak signals or missing values (e.g. the region labeled "2" in Fig. 3c).

### 2.2.2 Performance evaluation

The plume detection algorithm was applied to the $CO_2$ plumes of Berlin and Jänschwalde. Its performance was evaluated against the "true plume" defined by the field of the $CO_2$ tracer released by the respective source (CO2_BER in case of Berlin, CO2_PP for Jänschwalde) above a low threshold. For Jänschwalde, the true plume had to be additionally identified by visual inspection due to frequent overlaps with the plumes of other power plants also contained in the CO2_PP tracer.

To evaluate the performance of the Sentinel-5 with respect to detecting plumes as observed by CO2M, detected pixels had to be projected onto the pixels of the CO2M instrument. Sentinel-5 pixels were thus only used over the swath of the CO2M satellite rather than over the whole swath of Sentinel-5.

When using $NO_2$ or CO for plume detection, the performance was assessed by comparing the detected pixels with the true $CO_2$ plume rather than the true plume of the auxiliary gas. In this way, the degree of congruence between the $CO_2$ and auxiliary trace gas plumes was considered as well.

For the evaluation, we computed true positives (TP), false positives (FP) and the positive predictive value ($PPV = TP/(TP+FP)$) (Ting, 2010). A good algorithm should have a much smaller number of FP than TP and, therefore, a PPV close to 1.

## 3 Results

### 3.1 Coverage and potential for plume detection

In this section, the potential for plume detection is analyzed based on the simulated tracers emitted by the source of interest. These "true plumes" will be used in the following sections as reference to evaluate the performance of the plume detection algorithm. They can be interpreted as the maximum number of plumes detectable by a perfect, noise-free instrument.

The frequency with which the $CO_2$ plumes of a given source can be observed depends on how often a satellite passes over the source, how often the $CO_2$ signal is larger than the threshold and how often cloud-free conditions dominate during the overpass. We define an overpass as an intersection between the satellite swath and the "source region" as specified above. The number of overpasses scales with the number of satellites and the swath width of the instrument. The scaling with the number of satellites is not trivial, however, since individual satellites may pass over the source either once or twice during the 11-day repeat cycle depending on the satellite's equator starting longitude (see Fig. 2).




**Figure 5.** Examples of $XCO_2$ plumes of Berlin ($XCO_2$ signal > 0.05 ppm) with different cloud cover fractions (cc). The numbers of $XCO_2$ pixels are shown for cloud fractions ≤1%. (a-d) Plumes with increasing cloud fraction, (e) plume close to the edge of the swath, (f) plume without cloud-free $CO_2$ observations connected to Berlin.





The total number of overpasses per satellite is either 34 or 66 per year, depending on whether the satellite has one or two overpasses per 11-days repeat cycle. Since one out of four satellites has only one overpass per 11 days, the number of overpasses per year roughly scales with a factor 1.75 times the number of satellites times the number of 11-day periods per year (about 33). A constellation of six satellites covers the model domain nearly daily.

To define the extent of a plume in the satellite image, we have to set a signal threshold for the tracer field (XCO2_BER for Berlin) above which a pixel is considered as belonging to the plume. A possible threshold is the value at which the signal would become larger than the variability of the background, i.e. where the signal is larger than the standard deviation of the background. Based on the time series of standard deviations of the model background tracer (XCO2_BER_BG for $CO_2$) computed for a $200\times200$ km$^2$ square centered on the city of Berlin (Fig. 7c, f and i), we defined a threshold of 0.05 ppm for

$XCO_2$, $0.2\times10^{15}$ molec. cm$^{-2}$ for $NO_2$ and $0.06\times10^{17}$ molec. cm$^{-2}$ for CO approximately corresponding to the minimum of the standard deviations. Note that these thresholds are significantly smaller than the noise level of the instruments.

A plume was defined as the collection of pixels for which the signal is larger than the threshold. However, we also required that Berlin is inside the swath of the instrument to be able to unambiguously assign a plume to the city. Furthermore, we removed parts of plumes that re-entered the swath after leaving it, because it is often not possible to correctly assign these parts

to their source.

Since a satellite image can be obscured by clouds, we need to define how many pixels are needed to make up a "useful" plume. This number depends on the application. For example, to estimate emissions of cities, we require that the plume must extend beyond the city limits to contain emissions from the whole city area. The cross-wind diameter of Berlin's $CO_2$ plume is typically about 20 km or 10 satellite pixels, which is roughly the diameter of the part of the city with the highest emissions.

To cover at least the whole city area, we only consider $CO_2$ plumes with at least 100 cloud-free $CO_2$ pixels to be useful. For the power plants, the cross-wind diameter is less than five pixels near the source. Therefore, 10 pixels were used to define the minimum number for a useful plume in this case.

For a signal threshold of 0.05 ppm, Berlin's $CO_2$ plume has always more than 100 pixels above the signal threshold. However, many plumes are partly or fully covered by clouds significantly reducing the number of useful plumes. Figure 5a-d

presents examples of $CO_2$ plumes under different cloud conditions with an increasing fraction of cloudy pixels. Figures 5c and d show examples with plumes of only 11 and 20 pixels, much smaller than the area of the city. On the other hand, the 100-pixel threshold does not necessarily remove swaths with plumes in broken clouds (e.g. Fig. 5b), for which it will also be challenging to estimate emissions, because adjacent cloudy pixels increase the $XCO_2$ uncertainty (Taylor et al., 2016). The number of plumes with at least 100 pixels is also reduced when the source is close to the edge of the swath and winds are

pushing the plume out of the view of the satellite (e.g. Fig. 5e). These overpasses occur every 11 days due to the repeat cycle of the satellite. As a result, orbits with plumes near the edge of the swath can have up to 20% less useful plumes.

Figure 6 presents the number of useful city plumes (>100 pixels) per month for $CO_2$, $NO_2$ and CO for constellations of one to six satellites. Plumes without cloud-free observations over the source region (e.g. Fig. 5f) were removed, because they cannot be detected by the algorithm used in this study. A constellation of six satellites observes only $50\pm5$ $CO_2$ plumes within

one year despite almost daily overpasses due to the small number of days with low cloud fractions. Except for February, which

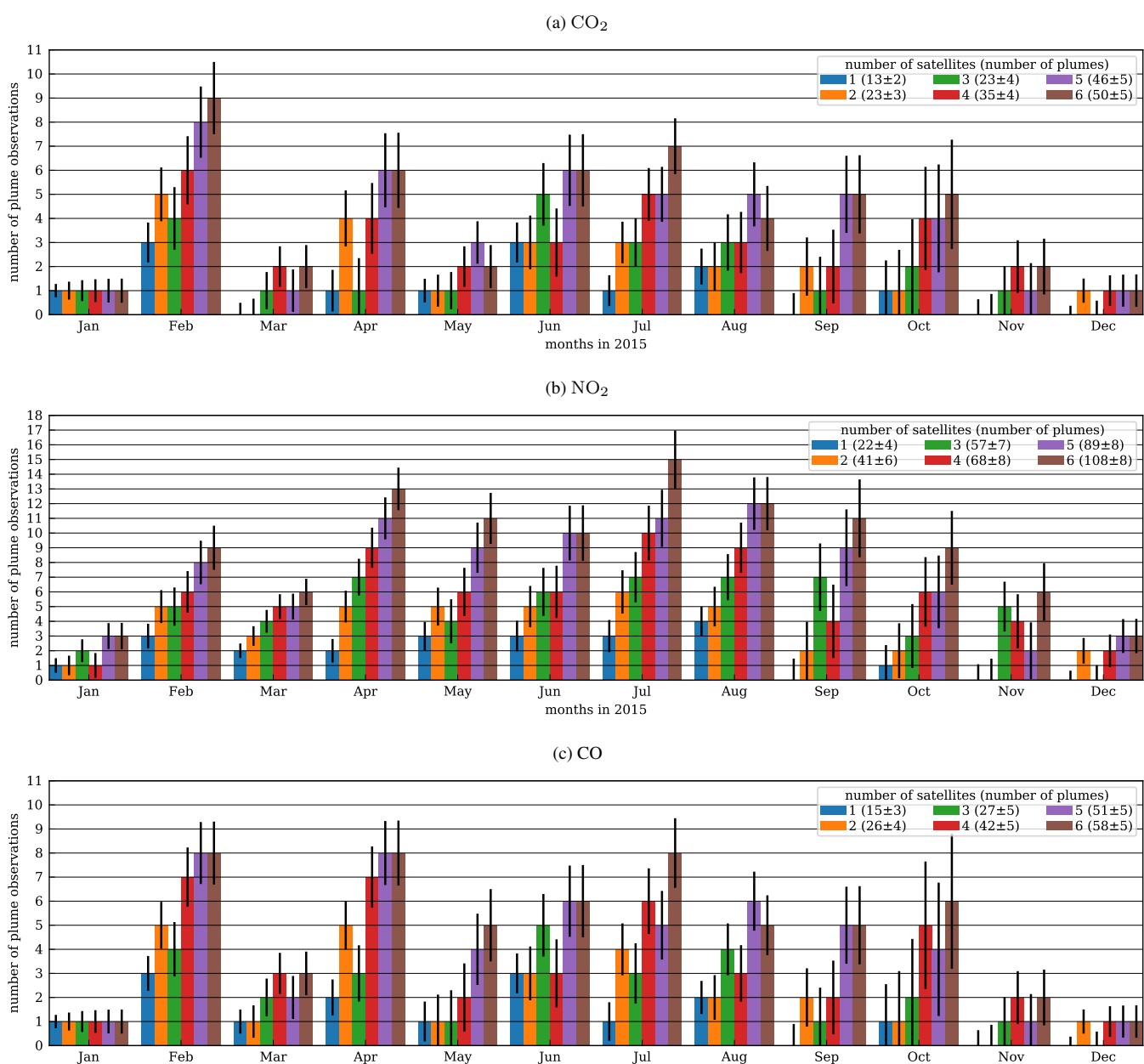

**Figure 6.** Number of cloud-free plumes of the city of Berlin with at least 100 pixels per month for (a) $CO_2$, (b) $NO_2$ and (c) CO. The cloud threshold is 1% for $CO_2$, 30% for $NO_2$ and 5% for CO observations. Error bars are obtained by comparing all available satellites. The number of expected plumes per satellite is 8, 17 and 9 for the $CO_2$, $NO_2$ and CO instrument, respectively.

was an unusually sunny month in 2015, there is a clear tendency of higher cloud fractions and correspondingly fewer plume observations in winter than in summer. The standard deviations shown in the figures as vertical black bars were estimated from the scatter of observable plumes using satellites with different equator starting longitudes. The presence of clouds thus reduces




**Table 5.** 5th, 50th and 95th percentile of $CO_2$, $NO_2$ and CO signals of Berlin as well as Jänschwalde and Lippendorf power stations.

| Species | 5th | 50th | 95th |
| --- | --- | --- | --- |
| Berlin: | | | |
| $CO_2$ (ppm) | 0.16 | 0.33 | 1.03 |
| $NO_2$ ($10^{16}$ molec. cm$^{-2}$) | 0.30 | 0.63 | 1.56 |
| CO ($10^{17}$ molec. cm$^{-2}$) | 0.12 | 0.27 | 0.96 |
| Jänschwalde power station: | | | |
| $CO_2$ (ppm) | 1.28 | 2.69 | 6.64 |
| $NO_2$ ($10^{16}$ molec. cm$^{-2}$) | 2.09 | 4.30 | 10.0 |
| CO ($10^{17}$ molec. cm$^{-2}$) | 0.55 | 1.14 | 2.88 |
| Lippendorf power station: | | | |
| $CO_2$ (ppm) | 0.53 | 1.29 | 3.73 |
| $NO_2$ ($10^{16}$ molec. cm$^{-2}$) | 0.85 | 2.03 | 5.37 |
| CO ($10^{17}$ molec. cm$^{-2}$) | 0.03 | 0.08 | 0.22 |

the opportunity for plume detection by a factor as a large as 5 to 6 over the city of Berlin. The number of observable $NO_2$ plumes per year (108±8) is about twice as large as for $CO_2$, which is primarily due to the larger cloud threshold of 30%. For CO the number of observable plumes per year is 58±5. The average number of plumes per satellite and year is thus about 8 (range: 3-13), 9 (4-15) and 17 (7-23) for $CO_2$, CO and $NO_2$, respectively.

The number of observable plumes varies strongly between the individual satellites of a constellation, because the number of cloud-free days per year is quite small and the overpass days are different for different equator starting longitudes. Since satellites are equally spaced in orbit, changing the number of satellites changes the starting longitudes and overpass days of the satellites. As a consequence, the number of observable plumes per constellation can also fluctuate strongly. According to Fig. 6, for example, a constellation of two satellites seems almost equivalent to a constellation of three, but this result is merely a consequence of the fact that cloud cover was often large during these overpasses and Berlin was at the edge of the swath for the satellite with a starting longitude of 8°. The result would be different for another starting longitude of the first satellite, another city, or another year.

### 3.2 Signal-to-noise ratios

The key measure that determines the detectability of a $CO_2$ plume is the SNR (Eq. 1), which compares the amplitude of the plume signal to the instrument noise and the variability of the background. SNRs provide a first indication of an instrument's suitability for detecting a plume.

Time series of the $CO_2$, $NO_2$ and CO plume signals were computed from the X_BER and X_PP tracers for Berlin and the power stations Jänschwalde and Lippendorf at the overpass time of CO2M, i.e. about 11 UTC. The signals were computed as maximum values of the local means within the source region, i.e. a circle with 15 or 5 km radius. Thereby, the local means were computed with a neighborhood $n_s$ of size 37 for Berlin and 5 for the power stations (Fig. 4). These sizes roughly correspond to







**Figure 7.** Time series of $CO_2$, $NO_2$, CO plume signals (left column), mean backgrounds (middle) and standard deviations of backgrounds (right) for Berlin. Signals are the largest local mean values of the X_BER model tracer using a 37-pixel-neighborhood. Background means and standard deviations were obtained from the X_BER_BG tracer for a $200 \times 200 \ \text{km}^2$ square centered over Berlin. Reference uncertainties ($\sigma_{\text{VEG50/ref}}$) corresponding to the different instrument scenarios are shown as horizontal lines for comparison.

the typical diameters of the $CO_2$ plumes from Berlin (about 15 km) and the power stations (about 6 km), respectively, and were also found most suitable for the plume detection algorithm (see also Kuhlmann et al., 2019). The results for Berlin are presented in Fig. 7 for the three trace gases. The figure compares the daily plume signals (left panels) to the daily mean background values (middle) and their spatial variability (right). The 5th, 50th and 95th percentiles of the time series are summarized in Table 5 for Berlin as well as for the power stations. The signals have a large range due to the variability of emissions (e.g. lower during weekends) and meteorology. The $CO_2$ and $NO_2$ signals of the power stations are between five and ten times larger than those of Berlin. The CO signal of Lippendorf, on the other hand, is smaller than the signal of Berlin. The power plants produce





**Table 6.** Median signal-to-noise ratios for signals of Berlin, Jänschwalde and Lippendorf using different uncertainty scenarios. The signals were computed as largest local mean values using a local neighborhood size $n_s$ of 37 and 5 for cities and power stations, respectively.

| Scenario name | Signal-to-noise ratio | | |
| --- | --- | --- | --- |
| | Berlin | Jänschwalde | Lippendorf |
| $CO_2$ low noise | 1.4 | 10.4 | 4.3 |
| $CO_2$ medium noise | 1.4 | 8.0 | 3.5 |
| $CO_2$ high noise | 1.2 | 5.8 | 2.6 |
| $NO_2$ low noise | 9.0 | 14.3 | 8.8 |
| $NO_2$ high noise | 8.4 | 10.8 | 7.5 |
| CO low/high noise | 0.4 | 0.6 | 0.0 |

strong local enhancements easily detectable by the $CO_2$ satellite, but the corresponding plumes are much narrower than those of Berlin.

Figures 7b and c present the spatial means and standard deviations of the background around Berlin. Background $XCO_2$ has a strong annual cycle with an amplitude of about 16 ppm. Since the $XCO_2$ plume signal of Berlin is typically only about

0.2 to 1.0 ppm, it is critical to accurately estimate the background $XCO_2$ value in Eq. (2). The spatial variability $\sigma_{bg}$ of the background, on the other hand, is typically only of the order of a few tenths of a ppm. Despite higher $XCO_2$ in winter than in summer, the variability is somewhat larger in summer due to stronger biospheric activity in combination with lower average wind speeds, especially in July and August. Large peaks in the background variability are often caused by plumes from other anthropogenic sources such as the power stations in the south-east of Berlin (Fig. 1).

For $NO_2$, the annual cycle of the background is relatively constant for our idealized $NO_2$ tracer with a constant lifetime of 4 hours (Fig. 7e and f). In reality, the lifetime will likely be longer and the variability correspondingly higher in winter. The $NO_2$ signal of Berlin is significantly larger than the background and its variability. Similar to $CO_2$, the CO time series has a strong annual cycle with an amplitude of about $5 \times 10^{17}$ molec. cm$^{-2}$ (Fig. 7h and i) requiring again an accurate estimation of the background. The standard deviation of the background is about half of the CO signal.

Table 6 summarizes the median of all SNRs of Berlin and the two power stations for the different satellite instrument scenarios that have been computed from the time series of highest signals. To understand the numbers, it should be noted that a plume pixel would be detectable when the SNR is larger than 2.3, i.e. $z(q) = 2.3$ for $q = 99\%$. For Berlin, the $CO_2$ SNRs are below this detection limit for all noise levels while $NO_2$ SNRs are above the limit. For the two power stations, SNRs are above the detection limit both for the $CO_2$ and $NO_2$ instrument scenarios, but SNRs for the $NO_2$ instrument scenarios are always

larger.

Based on the SNRs, the $NO_2$ plumes should be well detectable. For Berlin, the detection of the $CO_2$ plume with the $CO_2$ instrument will often be challenging due to low SNRs. The CO SNRs are always much smaller than those for $CO_2$, making a CO instrument with the given specifications little suitable for the purpose of plume detection. In the following, we therefore only investigate the potential benefit of auxiliary $NO_2$ observations.





### 3.3 Plume detection algorithm

The plume detection algorithm was applied to the $CO_2$ plumes of Berlin and Jänschwalde for different instrument scenarios. The probability $q$ was set to 99% and neighborhood sizes of 37 and 5 were selected for Berlin and the power stations, respectively. In the case of Sentinel-5, the corresponding neighborhood sizes were set to 5 and 1 due the larger pixels of this satellite.

Based on an analysis of the positive predictive values (PPV), these neighborhood sizes were found most suitable for detecting the city and power plant plumes (Kuhlmann et al., 2019). For Berlin, twenty synthetic satellite images were created for each single overpass with different patterns of random noise. The plume detection algorithm was subsequently applied to each image and the results were averaged to obtain more robust results independent of the selected noise pattern. For Jänschwalde, only one synthetic satellite image was created for each overpass, because no model tracer was available to compare the results with

a true plume.

#### 3.3.1 Examples of detected plumes from Berlin

Figure 8 shows the $CO_2$ and $NO_2$ plumes of Berlin on 21 April 2015 observed by CO2M and Sentinel-5 for different instrument scenarios. The outlines of the real plumes are overlaid as solid and dashed lines for $CO_2$ and $NO_2$, respectively. Since the $CO_2$ instruments have a lower cloud threshold, a band of cirrus clouds is obscuring the plume in the $CO_2$ observations but not

in $NO_2$. Successfully detected pixels are shown as black crosses and the number of detected pixels (median of all 20 noise realizations) are presented in the legend. On average, a $CO_2$ instrument detects 116±46, 48±40 and 24±30 pixels with noise scenarios $\sigma_{VEG50}$ of 0.5, 0.7 and 1.0 ppm, respectively (Fig. 8a and b). The number of true positives is slightly smaller having on average two false positive pixels. Consequently, the PPV is high, ranging between 0.85 and 0.99 for high and low noise, respectively.

For the $NO_2$ measurement the band of thin cirrus clouds is not an issue. The $NO_2$ instrument can therefore detect a much larger number of pixels, i.e., 1242±99 and 1203±155 in the case of the low and the high noise scenarios, respectively (Fig. 8c). On average, the fraction of FP is relatively large and the PPV is only 0.80±0.05 and 0.77±0.10 for the low and high noise scenarios, respectively. The small PPV is caused by interference with the plume of Jänschwalde, which is just south of the plume of Berlin. For cases where no neighboring plumes have been detected falsely with the $NO_2$ instrument, the spatial

match between $CO_2$ and $NO_2$ plumes is generally high, suggesting a high degree of spatial overlap between the $CO_2$ and $NO_2$ plumes.

The Sentinel-5 $NO_2$ instrument is also able to detect the $CO_2$ plume with 879±114 CO2M pixels, but since it is measured two hours earlier, the $NO_2$ plume seen by Sentinel-5 (dashed line in Fig. 8d) is slightly shifted with respect to the $CO_2$ plume (solid line). As a consequence, the PPVs is low (0.60±0.04).

Figure 9 presents two examples where the $CO_2$ instrument fails to detect the $CO_2$ plume. In Fig. 9a, the $CO_2$ field has a pronounced spatial gradient resulting in a high variance of the background. This gradient is not present in the much shorter lived trace gas $NO_2$ making it possible to detect the plume using an $NO_2$ instrument (Fig. 9b). Similar situations occur in roughly 20% of cloud-free swaths. Figures 9c and d show a second example where the $CO_2$ instrument cannot detect the



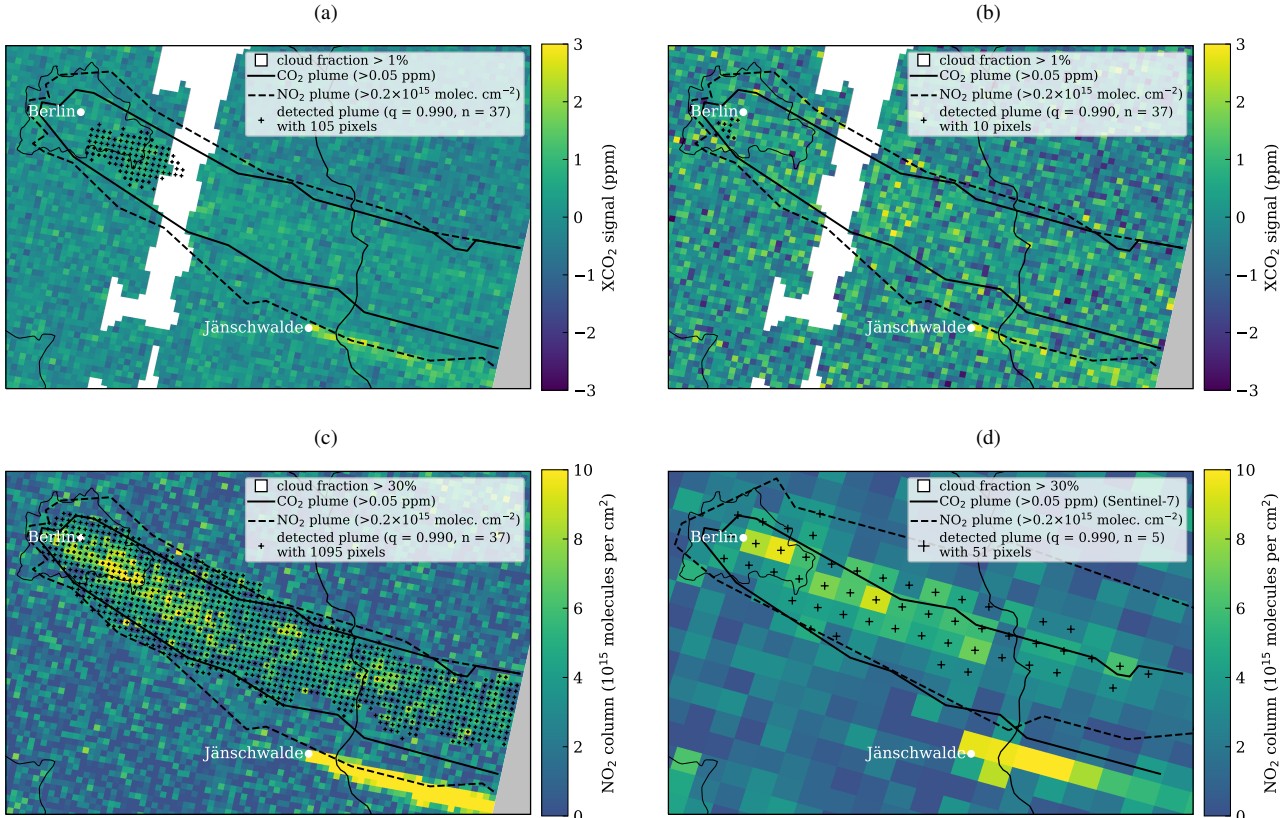

**Figure 8.** Example of plume detection with CO2M's $CO_2$ and $NO_2$ instrument and Sentinel-5's $NO_2$ instrument on 21 April 2015. Significant pixels detected by the algorithm are highlighted as black dots. The outlines of the true $CO_2$ and $NO_2$ plumes based on the $X_{BER}$ tracers are overlaid as solid and dashed lines, respectively. (a) Low-noise $CO_2$ instrument. (b) high-noise $CO_2$ instrument. (c) High-noise $NO_2$ instrument on the CO2M satellite. (d) $NO_2$ instrument on Sentinel-5.

plume, because the signal is very weak due to strong winds. Owing to its better SNR, the $NO_2$ instrument is able to detect the plume also in this situation.

Figure 10 presents two examples comparing the $NO_2$ plume observed by Sentinel-5 to the $CO_2$ plume observed two hours later by CO2M satellite. In the first example (panels a and b), Sentinel-5 fails to detect any plume due to clouds, which have
5  largely disappeared by the time of the CO2M overpass. In the second example, both Sentinel-5 and the CO2M satellite detect a plume of similar size, but the Sentinel-5 plume is significantly displaced due to changes in the prevailing winds between the two overpasses.

### 3.3.2 Number and sizes of detected Berlin plumes

To count the number of plumes detectable under the different instrument scenarios, we analyze the fifty plumes observed by a
10  constellation of six satellites, which we had classified in Sect. 3.1 as being potentially "useful" based on the idealized tracer

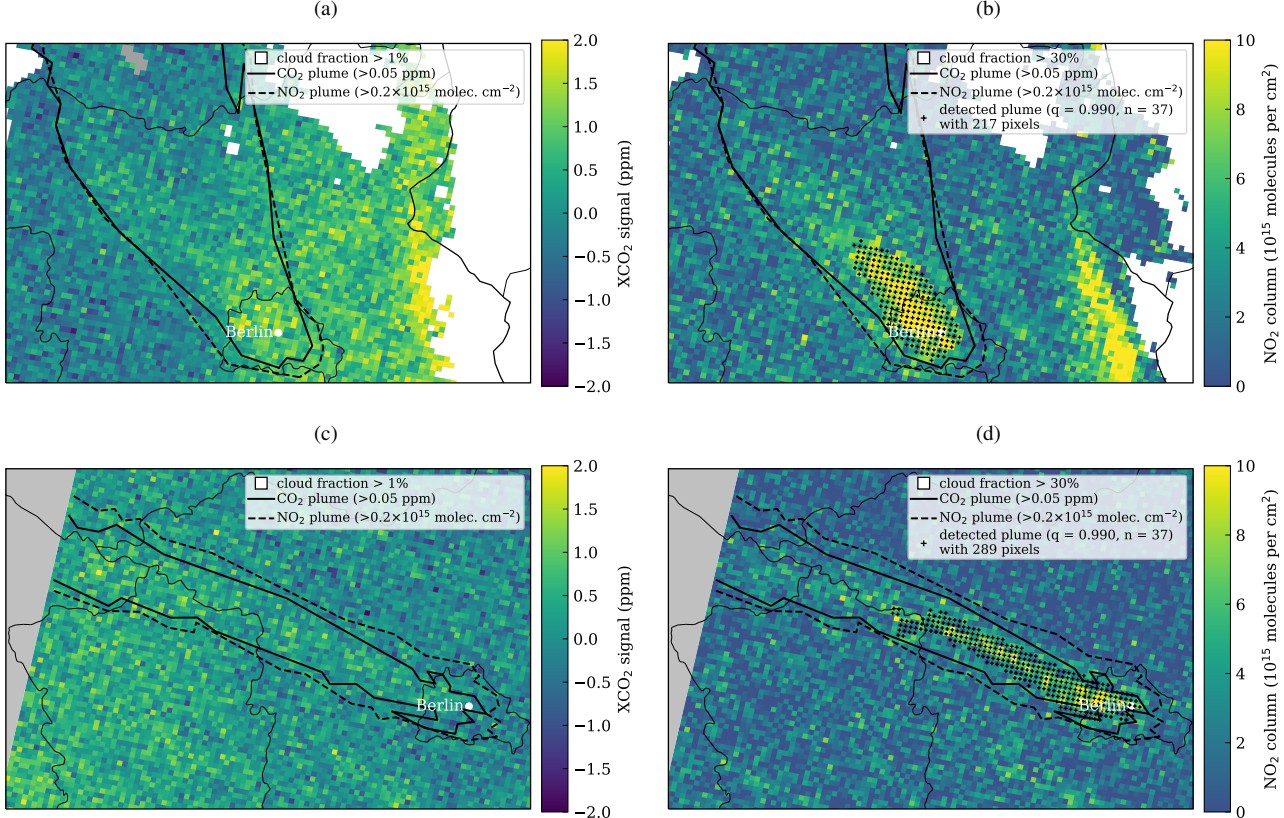

**Figure 9.** (a) Example of 27 February 2015 where plume detection with a $CO_2$ instrument fails because of pronounced horizontal gradients in the $CO_2$ background field. (c) Example of 2 July 2015 where the plume detection fails due to small $CO_2$ signals as a result of high wind speeds. In both cases, the plume can readily be detected with a high noise $NO_2$ instrument (panels b and d).

$X_{BER}$ having more than 100 pixels above a threshold of 0.05 ppm. Table 7 summarizes the results in terms of number and size of the detected plumes. A plume was only counted as detected when at least 100 $CO_2$ pixels were correctly detected (true positives) and when at least 80% of the detected pixels were true positives (PPV $\geq$ 0.80). The PPV threshold was found useful for removing plumes interfering with others or plumes shifted due to the earlier overpass time of Sentinel-5.

5    Table 7 shows that the $CO_2$ instruments detect significantly fewer plumes than the $NO_2$ instruments. Depending on instrument noise scenario, the $CO_2$ instruments detect plumes with more than 100 pixels with a success rate of only 12% to 32%, while for the $NO_2$ instruments the success rates are 68% to 70%. Surprisingly, the $NO_2$ instrument with low noise performs slightly worse than the high noise instrument. This is an artifact of the algorithm often detecting small plumes not related to emissions from Berlin in the case of a low-noise instrument. The Sentinel-5 $NO_2$ instrument detects 20% of the plumes, thus

10    only half the success rate of the $NO_2$ instrument on the CO2M satellite. The main reason for this low success rate is the spatial mismatch of the plumes due to the two hours difference in overpass times.

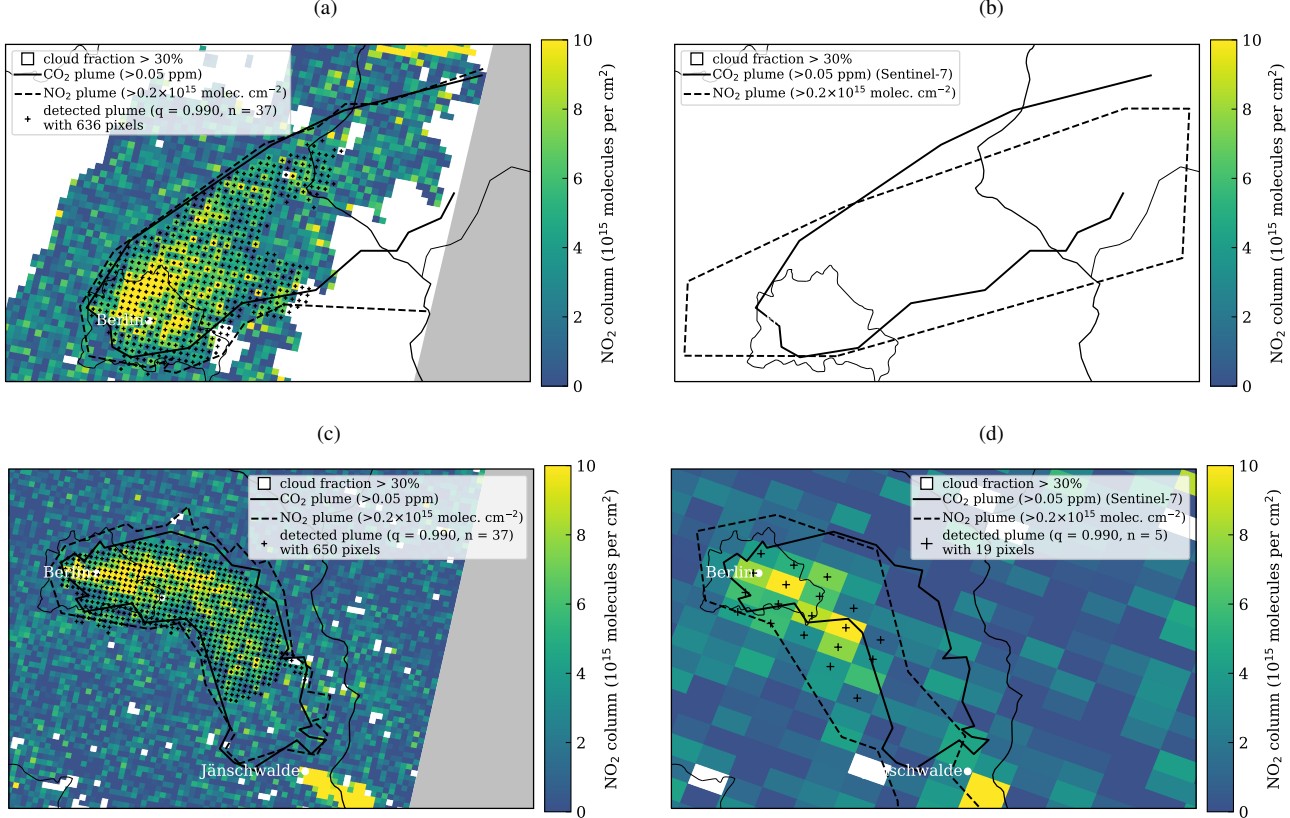

**Figure 10.** Examples of comparing plume detection between the Sentinel-5 and CO2M's $NO_2$ instruments (a, b) Due to time-lag of two hours, the scene is cloudy during the Sentinel-5 overpass but not during the CO2M overpass (3 December 2015). (c, d) The plume position clearly changed between the two overpasses (17 June 2015).

**Table 7.** Number of $CO_2$ plumes from Berlin and 5th, 50th and 95th percentiles of the number of detected $CO_2$ pixels (TP if PPV $\geq$ 0.8 and cloud fraction < 1%) for a constellation of six CO2M satellites or one Sentinel-5 satellite. The maximum number of detectable plumes would be 50, corresponding to all potentially useful plumes with at least 100 $CO_2$ pixels with values of the tracer $X_{BER}$ above a threshold of 0.05 ppm.

| Instrument scenario | Plumes with $\geq$100 $CO_2$ pixels | | Plume size at percentile | | |
|---|---|---|---|---|---|
| | number | percentage (%) | 5th | 50th | 95th |
| $CO_2$ low noise | 16±1 | 32±3 | 0 | 5 | 323 |
| $CO_2$ medium noise | 10±1 | 20±3 | 0 | 3 | 261 |
| $CO_2$ high noise | 6±1 | 12±2 | 0 | 7 | 181 |
| $NO_2$ low noise | 34±1 | 68±2 | 52 | 294 | 600 |
| $NO_2$ high noise | 35±2 | 70±3 | 50 | 279 | 527 |
| $NO_2$ Sentinel-5 | 10±2 | 20±3 | 0 | 140 | 396 |

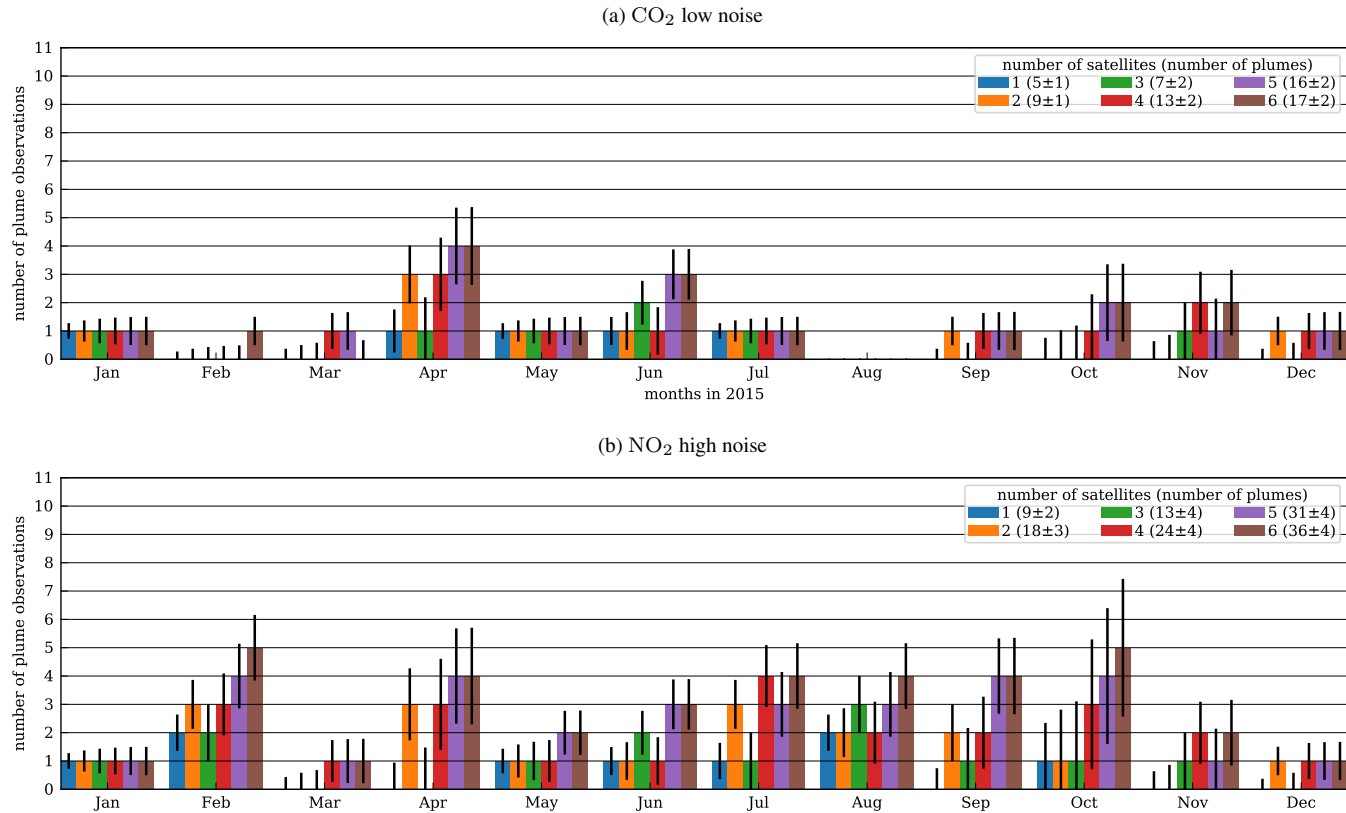

**Figure 11.** Number of detected plumes with at least 100 pixels (TP≥100 and PPV≥0.80) for one to six satellites with the (a) $CO_2$ low noise and (b) $NO_2$ high noise scenario. Error bars were estimated from all available satellites with different equator starting longitudes.

Figure 11 shows the number of plumes per month with at least 100 detected pixels for different constellations between one and six satellites for the $CO_2$ low noise and the $NO_2$ high noise scenario. The number of plume detections per month is small and therefore highly sensitive to the specific orbit configuration. For example, two satellites seem to detect more plumes than three, but this result is caused by an unfavorable orbit for observing Berlin for the constellation with three satellites and

5   unfavorable cloud cover as already discussed earlier. The standard deviation was estimated from the number of detectable plumes using satellites with different starting longitudes (i.e. east-west displacements of all orbits). The figure shows that the number of observed plumes generally increases with the number of satellites as expected, but statistical noise can mask the increase from one constellation to the next. The figure confirms the much lower success rates of the $CO_2$ instruments as compared to the $NO_2$ instruments as expected from the computed signal-to-noise ratios.

10  **3.3.3   Detection of plumes from power stations**

There are six major power plants in the model domain: Jänschwalde, Boxberg, Schwarze Pumpe, Lippendorf, Turow and Patnow. Because no model tracer was defined for individual power plants but only for the sum of all of them, the true plume







**Figure 12.** Example of plume detection for Jänschwalde power plant on 2 November 2015 using (a, b) the $CO_2$ instrument with $\sigma_{VEG50}$ of 0.5 and 1.0 ppm, (c) the $NO_2$ instrument with the high noise scenario and (d) the $NO_2$ instrument on Sentinel-5.

of an individual power plant is not known. Therefore, we applied a visual inspection to identify those plume detections which erroneously included neighboring plumes. Furthermore, we limit the analysis to Jänschwalde.

As an example, Fig. 12 shows the successful detection of the $CO_2$ plume of Jänschwalde on 2 November 2015 by different instruments. Since $CO_2$ emissions of Jänschwalde are high with 33.3 Mt $CO_2$ yr$^{-1}$, the $XCO_2$ signal is very strong and can be detected well even with a high noise instrument ($\sigma_{VEG50} = 1.0$ ppm). With low noise ($\sigma_{VEG50} = 0.5$ ppm) the weaker plumes of Schwarze Pumpe and Boxberg are visible as well. The $NO_2$ instrument detects the four plumes in the region well. On this day also the Sentinel-5 $NO_2$ instrument successfully detects the plume of Jänschwalde and other point sources. Figure 13 presents





**Figure 13.** Example of plume detection for Jänschwalde power plant on 17 February 2015 using (a, b) the $CO_2$ instrument with $\sigma_{VEG50}$ of 0.5 and 1.0 ppm, (c) the $NO_2$ instrument with the high noise scenario and (d) the $NO_2$ instrument on Sentinel-5





**Table 8.** Number of plumes detected for Jänschwalde with six satellites. The number of plumes are provided for plumes with at least 10 detected $CO_2$ or $NO_2$ pixels and in addition with at least 10 $CO_2$ pixels (cloud cover <1%). In both cases, plumes were included where neighboring plumes were detected in addition to Jänschwalde, i.e. detection with a large number of false positves. These plumes are also shown separately. The classification uncertainty is about ±5 plumes. The neighborhood size was set to $n_s = 5$.

| Instrument scenario | Number of plumes with | | |
|---|---|---|---|
| | $\geq 10$ detected pixels | $\geq 10$ $CO_2$ pixels | large number of false positives |
| $CO_2$ low noise | 44 | 42 | 7 |
| $CO_2$ medium noise | 42 | 40 | 6 |
| $CO_2$ high noise | 41 | 40 | 4 |
| $NO_2$ low noise | 90 | 68 | 38 |
| $NO_2$ high noise | 91 | 68 | 34 |

a second, more challenging example for 17 February 2015. The $CO_2$ instrument successfully detects the plume with 0.5 ppm uncertainty, but with 1.0 ppm uncertainty, the number of detected pixels is likely too small to be useful for emission estimation. The reason for the low number of detected pixels in this case is the strong horizontal gradient in the $CO_2$ background. The $NO_2$ instrument detects the plume, but because the $NO_2$ plume of Jänschwalde overlaps with neighboring plumes, these plumes are

erroneously assigned to Jänschwalde as well. At the coarser resolution of Sentinel-5 the plumes of the individual power plants can hardly be separated and, moreover, the time difference of two hours results in a plume location that is shifted with respect to the plume seen by the CO2M satellite.

    Table 8 summarizes the results of the plume detection for Jänschwalde under the different instrument scenarios. It shows the number of detected plumes with at least 10 pixels and in addition, the number of plumes with at least 10 valid $CO_2$ pixels (cloud

cover <1%). Note that in the case of the much narrower plumes from power plants, fewer pixels are required to form a useful plume. We classified detections that include large parts of the background as failed, but still counted detections that include neighboring plumes as successful, because they successfully identified the location of the plume. Since the classification is not always unambiguous, we assigned an uncertainty of about ±5 plumes at most.

    In the year 2015, the number of detectable plumes with more than 10 pixels for a constellation of six satellites was between

40 and 45 for a $CO_2$ instrument with $\sigma_{VEG50}$ of 0.5, 0.7 and 1.0 ppm. At the same time, the $NO_2$ instrument detected about 90 plumes for the low and high noise scenario. When only plumes with more than 10 cloud-free $CO_2$ observations were considered, the number was reduced to about 70 plumes. For a smaller number of satellites, the number of detectable plumes would be correspondingly smaller. The $NO_2$ instrument detects more plumes because of its lower sensitivity to clouds, which makes it possible to trace the plume to the source even for partly cloudy scenes. On the other hand, the $NO_2$ instrument

often detects overlapping plumes (e.g. Boxberg and Schwarze Pumpe), because the instrument is much more sensitive to small signals further away from the origin than the $CO_2$ instrument. The mean plume size was about 100 pixels for the low noise $CO_2$ instrument. The plumes detected with the high noise $CO_2$ instrument were about half the size. The $NO_2$ instruments detected a similar number of $CO_2$ pixels as the low noise $CO_2$ instrument, but when all detected pixels are counted the number of pixels doubles.





## 4  Discussion

### 4.1  Comparison with previous studies

In this study we investigated whether and how frequently the $CO_2$ plume of Berlin and power stations can be detected by different constellations of satellites using either $CO_2$ observations alone or in combination with observations of the co-emitted

trace gases CO and $NO_2$. To address the question, high-resolution $CO_2$, CO and $NO_2$ fields were simulated with the COSMO-GHG model for the year 2015 and used to generate synthetic $XCO_2$, CO and $NO_2$ satellite observations for Sentinel-5 and a constellation of CO2M satellites. Similar OSSEs studies were conducted by Pillai et al. (2016) and Broquet et al. (2018) for Berlin and Paris, respectively, as part of the LOGOFLUX study (Bacour et al., 2015). However, their simulations did not include $NO_2$ and CO fields. A fundamental difference is the realistic, i.e. not as a random noise, account for transport model

errors in the SMARTCARB study, where the location of the plume is not taken from the model but detected in the satellite image using either $CO_2$ or $NO_2$ observations. For this reason, the focus in this paper is on the detectability of the plume, while Pillai et al. (2016) and Broquet et al. (2018) focused on the inversion, which we describe in another publication.

   Pillai et al. (2016) simulated $CO_2$ fields with the WRF-GHG model with $10 \times 10$ km$^2$ spatial resolution for the year 2008. The resolution was relatively low compared to the $1.1 \times 1.1$ km$^2$ resolution used our study. They used $CO_2$ emissions from

the EDGAR inventory (Version 4.1), which are more than twice as high as the emissions reported in the inventory of the city of Berlin as mentioned earlier. A consequence of the unrealistic high emissions are higher $CO_2$ signals (0.80 - 1.35 ppm) for Berlin than in our study (0.16 - 1.03 ppm, see Table 5). Note that the signal strength also depends on the spatial resolution, but our $XCO_2$ signals were computed for a local mean ($n_s$ = 37, i.e. 148 km$^2$ spatial resolution) that is comparable to the model resolution used by Pillai et al. (2016). For Paris, Broquet et al. (2018) conducted simulations with the CHIMERE atmospheric

transport model with 2 km spatial resolution. $CO_2$ emissions for the greater urban area were 40-50 Mt $CO_2$ yr$^{-1}$ that resulted in a $XCO_2$ signal of ~1 ppm, quite consistent with the plume signals reported here.

   For Berlin, we estimated that 3 to 13 "useful" plumes (defined as plumes with at least 100 pixels above a threshold of 0.05 ppm) would be observable during one year by a single Sentinel-$CO_2$ satellite with a 250-km wide swath. Pillai et al. (2016) identified 17 and 27 plumes with a 240 km and 500 km swath, respectively. Their number is larger than the maximum

value for the range found in our study, partly because small plumes with less than 100 pixels were excluded in our case. The remaining difference is likely explained by their using higher emissions and not considering vertical profiles of emissions, which results in stronger and correspondingly larger plumes (Brunner et al., 2019). Furthermore, differences are to be expected due to different meteorology, especially cloud cover and wind speed, for the different simulation periods.

   Since the number of "useful" plumes observable by a satellite critically depends on its orbit (with one or two overpasses

over Berlin per 11-day repeat cycle), the number of useful plumes per satellite may easily be overestimated if only an optimal orbit is considered. Comparing different equator starting longitudes as applied in this study reduces the sensitivity to a specific orbit selection.





## 4.2 Benefits of CO and NO$_2$ measurements

Out of 50 potentially useful plumes, many plumes were too weak to be easily seen by the CO$_2$ instrument. Plumes of Berlin with more than 100 detectable pixels could only be detected in about 12% of cloud-free cases with a high-noise instrument ($\sigma_{\mathrm{VEG50}} = 1.0$ ppm) and about 32% with a low-noise instrument ($\sigma_{\mathrm{VEG50}} = 0.5$ ppm). The success rate of 32% for an imperfect but still precise instrument ($\sigma_{\mathrm{VEG50}} = 0.5$ ppm) would only allow for 2 to 3 favorable plume observations per year and satellite. These numbers illustrate the challenge and call for a larger constellation or a wider swath to increase the opportunities for plume detection and emission quantification, and for a CO$_2$ instrument with as low noise as possible.

Adding an NO$_2$ instrument greatly enhanced the opportunities for detecting the CO$_2$ plumes (68%-70% of cloud-free cases), since the NO$_2$ plumes largely overlap with the CO$_2$ plumes and since the signal-to-noise ratio is better for the NO$_2$ instrument. Furthermore, variability in the background is less important than for CO$_2$, and the NO$_2$ observations are less sensitive to clouds. Nevertheless, even with an NO$_2$ instrument, the number of detectable plumes per satellite remained small, of the order of 5 to 6 per year. A CO instrument with specifications similar to the CO instrument on Sentinel-5 had a smaller signal-to-noise ratio than the CO$_2$ instrument. Such an instrument would add little useful information over a developed region like Germany where combustion processes are well controlled and CO:CO$_2$ emission ratios correspondingly small. The question whether CO signals would be sufficiently high in other regions of the globe was outside the scope of this study.

The Sentinel-5 NO$_2$ instrument is well suited to detect the NO$_2$ plume of Berlin. However, the different overpass times of the CO2M satellite (11:30 local time) and Sentinel-5 (9:30 local time) frequently resulted in a significant spatial mismatch between the plumes, which reduced the number of matching plumes to 20% of the cloud-free cases. This is similar as for the CO$_2$ instrument with medium noise but three to four times lower than in the case of an NO$_2$ instrument placed directly on the CO2M satellite.

The detection of plumes from strong point sources like the power plant Jänschwalde was easier than the detection of city plumes, because point sources tend to have stronger and more confined CO$_2$ plumes for the same amount of emitted CO$_2$. In addition, the number of pixels required to map out such a plume was smaller, with only 10 detectable pixels being typically sufficient for identifying the main part of the plume. With a constellation of six satellites, about 40-45 plumes from Jänschwalde (33.3 Mt CO$_2$ yr$^{-1}$ emissions) with more than 10 detectable pixels could be observed per year even with a high-noise CO$_2$ instrument. This corresponds to 6 to 8 plumes per satellite and year, which is significantly better than for Berlin with the best instrument. The number of detectable plumes further increased by about 50% with the NO$_2$ instrument (about 70 plumes). Smaller point sources with emissions of about 10 Mt CO$_2$ yr$^{-1}$ (e.g. Lippendorf, Schwarze Pumpe and Turow) were sometimes detectable with a low-noise CO$_2$ instrument, but could also be detected with an NO$_2$ instrument.

Our study did not include systematic errors in the satellite observations, which can result in spatial patterns resembling plume structures and therefore complicate plume detection. Systematic errors affect all satellite products, but the effect would be more severe for XCO$_2$ than for NO$_2$ due to the much smaller signal-to-noise ratios.





Since the shape and extent of the plume can be imaged more accurately with an $NO_2$ instrument on the CO2M satellites, the $NO_2$ instrument can also be used to assess and correct transport simulations and improve these simulations through data assimilation.

### 4.3 Limitations of current plume detection algorithm

The plume detection algorithm presented here worked well even with weak signals well below the single sounding precision, but tended to fail when the $CO_2$ or $NO_2$ field was complex, for example when several plumes from adjacent sources overlapped or when the background had a spatial gradient. These cases can be easily identified by a trained human as done in this study, but will have to be automatized for application at the global scale, for example by applying machine learning methods.

The algorithm assumed accurate knowledge of the mean and variance of the background, which in a more realistic setup

would have to be estimated directly from the satellite observations. It is possible that due to this optimistic assumption the number of detectable plumes was overestimates. On the other hand, mean and variance were computed in a rather simple way from a large window centered on the source ($200 \times 200$ km$^2$ for Berlin). In the present algorithm, spatial gradients in the background field contributed to the variance of the background and thus reduced the ability for plume detection. However, such gradients, if sufficiently smooth, could potentially be accounted for in a more advanced algorithm through spatial interpolation

of the background surrounding the plume.

Because atmospheric transport models have large uncertainties at the scale of individual plumes, a data-driven approach to plume detection is essential. Nevertheless, a well-considered inclusion of model data, climatologies and other measurements could help constrain and detect the location of the plume, for example, by estimating the a priori plume location based on wind direction and speed, by estimating mean and variance of the background, and by adjusting the size of the neighborhoods based

on plume type (city or point source) and wind speed. Further improvements of the algorithm presented here have the potential for increasing the number of detectable plumes per satellite as well as the number of $CO_2$ pixels per plume.

### 5  Conclusions

In this paper the potential for detecting $CO_2$ plumes of the city of Berlin and neighboring power stations was investigated for the Copernicus anthropogenic $CO_2$ monitoring mission (CO2M), which is a proposed constellation of $CO_2$ satellites of the

European Copernicus program. Since the interference of biospheric $CO_2$ makes the identification of weak anthropogenic $CO_2$ plumes challenging, plumes were detected either from $CO_2$ observations or from observations of the co-emitted trace gases CO and $NO_2$. The study used high-resolution atmospheric transport simulations to create realistic $CO_2$, CO and $NO_2$ fields at $1 \times 1$ km$^2$ horizontal resolution to generate synthetic observations of $XCO_2$, CO and $NO_2$ for constellations of up to six CO2M satellites and one Sentinel-5 satellite.

For the city of Berlin about $50 \pm 5$ potentially "useful" $CO_2$ plumes were identified for the year 2015 for a constellation of six satellites, i.e. about eight plumes could be observed by a single CO2M satellite per year. This number is somewhat smaller than reported in earlier studies (Bacour et al., 2015; Pillai et al., 2016), but it is consistent because small plumes with less than 100





pixels were excluded in our case. Many of these fifty potentially observable plumes were too weak to be easily detectable by the $CO_2$ instrument. Plumes with more than 100 detectable pixels could only be identified in 12% and 32% of cloud-free cases with a high-noise ($\sigma_{VEG50}$ = 1.0 ppm) and low-noise $CO_2$ instrument ($\sigma_{VEG50}$ = 0.5 ppm), respectively. A CO instrument with the uncertainty scenario used in this study had a signal-to-noise ratio that was lower than for the $CO_2$ instrument and

was therefore not suitable for detecting $CO_2$ plumes. On the other hand, adding an $NO_2$ instrument significantly increased the number of detectable plumes (68%-70% of cloud-free cases), because $CO_2$ and $NO_2$ plumes generally overlapped well. The better performance of the $NO_2$ instrument was partly due to the higher signal-to-noise ratio and partly due to the lower sensitivity to clouds. The Sentinel-5 instrument was also well suited to detect the $NO_2$ plumes, but the different overpass times of the CO2M satellites (11:30 local time) and Sentinel-5 (9:30 local time) often resulted in a large spatial mismatch.

Strong point sources like the power plant Jänschwalde could be detected more easily with the $CO_2$ instrument (40-45 plumes), because the plumes were spatially more confined and the signals were stronger. The number of detectable plumes increased further with the $NO_2$ instrument by about 50% (about 70 plumes). The Sentinel-5 instrument could also detect the $CO_2$ plume, but could not always distinguish the plumes from neighboring power plants due to the lower spatial resolution. In addition, the spatial mismatch between CO2M and Sentinel-5 was large due to the 2-hour time difference between overpasses.

Smaller point sources with emissions of about 10 Mt $CO_2$ yr$^{-1}$ were only detectable with a low-noise $CO_2$ instrument, but were in most cases readily detectable with an $NO_2$ instrument.

In this study, power plant plumes could be detected even with an $NO_2$ instrument with high noise. The power plants were equipped with wet scrubber technology for reducing $SO_2$ and $NO_x$ emissions but not with the latest available technology. Future updates using selective catalytic or non-catalytic reduction have the potential to further reduce $NO_x$ emissions by 20

to 50% (Lecomte et al., 2017), which would place higher requirements on the $NO_2$ instrument and would make the low noise scenario more beneficial.

This study demonstrates the huge benefit of adding an $NO_2$ instrument to a constellation of CO2M satellites for detecting city plumes and weaker point sources. The major advantages of the $NO_2$ instruments are the higher signal-to-noise ratio and the lower sensitivity to clouds. Therefore, adding an $NO_2$ instrument is highly recommended and the low-noise instrument is

preferable for detecting also weaker plumes. Furthermore, development of an advanced plume detection algorithms that can detect $CO_2$ plumes reliable will be essential for the application on an operational satellite.

*Code and data availability.* Column averaged dry air mole fractions of all simulated tracers are available both as 2D fields and as synthetic satellite products through ESA. The total 3-dimensional model output amounts to 7.5 TB and is archived at Empa servers. Selected fields or time periods can be made available upon request. The TNO/MACC-3 inventory is available through Copernicus (http://macc.copernicus-

atmosphere.eu). The emissions inventory of Berlin was kindly provided by Andreas Kerschbaumer, Senatsverwaltung Berlin, and is available for research upon request. The global $CO_2$ simulation, which provided the lateral boundary conditions, was conducted in the framework of the European Earth observation program Copernicus and can be retrieved from ECMWF's MARS archive as experiment gvri, stream lwda, class rd.



*Author contributions.* GK designed the model experiments, conducted the simulations, and wrote the manuscript with input from all co-authors. GB followed the project as external advisor and contributed critical input to the manuscript. JM contributed the VPRM flux data required for the simulations. VC ported the COSMO-GHG extension to GPUs. YM and AL accompanied the study as ESA project and technical officers, respectively, and provided critical inputs and reviews during all phases of the project. DB led the SMARTCARB project
and contributed critical input to manuscript.

*Competing interests.* The authors declare that they have no conflict of interest.

*Acknowledgements.* This study was conducted in the context of the project SMARTCARB funded by the European Space Agency (ESA) under contract no. 4000119599/16/NL/FF/mg. The views expressed here can in no way be taken to reflect the official opinion of ESA. The work was supported by a grant from the Swiss National Supercomputing Centre (CSCS) under project ID d73. We acknowledge the
contributions of the Federal Office for Meteorology and Climatology (MeteoSwiss), the Swiss National Supercomputing Centre (CSCS), and ETH Zurich to the development of the GPU-accelerated version of COSMO. We like to thank Oliver Fuhrer (Meteoswiss) who supported the porting of COSMO-GHG to GPUs and the setup of the model on the supercomputer at CSCS. We would like to acknowledge Richard Engelen and Anna Agusti-Pannareda (ECMWF) as well as the European Earth Observation Program Copernicus and the EU project CHE for providing support and access to global $CO_2$ model simulation fields. The TNO/MACC-3 emissions inventory and temporal emission profiles
were kindly provided by Hugo Denier van der Gon (TNO, The Netherlands). We also like to thank Jochen Landgraf and Joost aan de Brugh (SRON) for providing the orbit simulator. We are also very grateful to Andreas Kerschbaumer, Senatsverwaltung Berlin, for providing the emission inventory of Berlin and additional material and for being available for discussions on its proper usage.





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
