# Peer review of "Detectability of CO2 emission plumes of cities and power plants with the Copernicus Anthropogenic CO2 Monitoring (CO2M) mission"

_Atmospheric Measurement Techniques, 2019_

## Referee Comment (RC1) · Anonymous Referee #1 · 19 Jul 2019

The manuscript describes a method for the detection of CO2 plumes of point and city-scale sources. The authors apply the developed method to simulated satellite data of the envisaged CO2M Copernicus mission in different constellations with 1 to 6 satellites. By the example of Berlin and nearby power plants, they derive the number of expected detectable plumes potentially useful for flux retrievals and assess the benefits of simultaneous or S5 NO2 and CO retrievals. Overall, the manuscript is well structured and illustrated with figures of good quality. The topic of the manuscript is relevant for future emission monitoring activities and fits well in the aims and scopes of AMT. Nevertheless, I have several comments and I would recommend a publication only after a corresponding revision of the manuscript.

[Figure]

**1  General comments**

The proposed method cannot be applied to real satellite data as it is now, because a key problem is not addressed, namely how to derive the background and its uncertainty. This should be discussed more prominently also in the conclusions and already be mentioned in the abstract.

To me, many points in the description of the algorithm are unclear (see my specific comments).

$NO_2$ can be used in more cloud contaminated scenes compared to $CO_2$. This is one reason why you find more plumes in $NO_2$. When computing the number of successfully found inner-plume soundings with $NO_2$, do you apply the stricter $CO_2$ cloud filter in the last step? This would be very important, because only these soundings could be used for the flux estimation. Please clarify this in the manuscript and if this last filter is not applied, discuss why you think that your results are still meaningful for CO2M. Which effect is dominating the advantages of $NO_2$, less strict cloud filtering or better SNR? How would both compare if $CO_2$ cloud filtering would have been also applied to $NO_2$?

In respect to a potential future application to real data, the pros and cons of plume detection based flux retrieval methods should be compared to inverse modeling approaches in more depth. Some examples: Inverse modeling can be applied also to plumes below the detection limit. Inverse modelling may place the plumes wrongly so that modelled plumes become less correlated with measured plumes resulting in underestimated fluxes. Plume detection depends on plume strength, so that this may introduce an observational bias towards large fluxes, i.e., annual average fluxes will likely be biased high, so that comparisons of reported and measured fluxes are more meaningful on a per-overflight basis. Plume detection methods can be applied to unknown source positions. Etc.

Why do you consider PPV a good measure for performance? As an example: consider

a very conservative plume detection algorithm, detecting only soundings which are 20ppm larger than the BG. In this case, TP would admittedly become small, but FP would become zero, so that PPV would become 1. Could the Hanssen-Kuiper Skill Score or True Skill Statistic be a better option?

To my knowledge, the Z-test is appropriate only for samples with more than 30 elements. 5 out of 8 tested smoothing kernels (Fig.4) consist of less elements. Why do you think the z-test is still appropriate in these cases?

What happens at cloud edges and at the edges of the swath? Effectively, the smoothing kernel here has fewer elements. Is this considered?

**2  Specific comments**

P2L23: Please also cite Reuter et al. 2014 in this context. They analyzed $CO_2$+$NO_2$ measurements from the same sensor (SCIAMACHY) and proposed multi-species measurements for future satellite missions. As signals in SCIAMACHY data are smaller because of the large pixel size, they had to follow a statistical approach. Note that this does not only cover strong localized sources, but all localized sources.

P3L4-L5: I would suggest moving this outlook to the conclusions section.

P3L8-L27: Strictly speaking, everything in these lines does neither belong to "data" nor to methods. Maybe it would fit better in the introduction.

P3L24: The situation is also not that simple for individual point sources, because chemistry changes the $NO_2$ plume which is not the case for the $CO_2$ plume. The following points should be addressed in this context: 1) $NO_2$ decays with time. 2) $NO_2$ and $CO_2$ can be emitted/transported into various altitudes with differences chemical regimes. For example, the high altitude part of the plume may decay faster/slower compared with the low altitude part of the plume. As the wind direction often changes with height, the

satellite observed CO2 plume may have a different (wider) shape as the NO2 plume. 3) NO2 is primarily emitted as NO (which later forms NO2), which can result in different plume shapes especially near the source.

P7L4: sun-synchronous orbits are not circular.

P7L5: "... in case of 3 satellites." Please describe what this means (the satellites will have the same Eq. crossing times...).

P9L30: Cloud coverage is a main driver for data yield of XCO2 satellite products. I guess COSMO diagnoses fractional cloud coverage in its 3D model domain. How do you/COSMO compute the 2D cloud fractional coverage from this (e.g., maximum/minimum overlap assumption) and how realistic are the 2D cloud coverage statistics for this particular model?

P10 Fig3 Caption: "... are marked with "T" for True." Shouldn't be True = blue/red and T = detected by Z-test? If so, consider replacing T by Z.

P11L7: The choice of neighborhood size appears a bit arbitrary. Please discuss, why it makes sense to try neighborhood sizes as selected. For example, I could imagine, that the maximum meaningful neighborhood size is reached once the noise of the smoothed pixels dropped significantly below levels of typical enhancements. On the other hand, the neighborhood should probably never be significantly larger than the typical plume extends.

P11L20,L21: Please describe why the Z-test calculates the SNR (e.g., the variance of the basic population is assumed to be known from the measurement erros, $X_{obs is an estimator for the expectation value}$).

P11Eq1: On P9L4 you stated that systematic errors are not scope of this study. Why considering them in Eq1? Xobs are the smoothed observations (not to be confused with the original observations). This should be corrected also at P11L13. $Sigma_{r and is only approximately the same for all soundings within the smoothing kernel but strictly speaking, it is different for}$

P11Eq2: Does this imply that there will also be false positives just by chance? How often is this expected to happen?

P11L30: Does this mean sigma$_s ys = sigma_b g in Eq.1$?

P12L1: What is the chosen level for significance?

P12L5: Please explain "Moor neighborhood".

P12L9: "... radius of 5km" In reality, this can result in non-detected NO2 plumes when NO2 is emitted primarily as NO.

P12L21: Can you separate both effects; which one usually dominates? In other words, why is a plume not detected, too low signal or CO2 plume != NO2 plume?

P13Fig.5d: How can you compute X$_o bs in totally cloudy regimes?$

P14L14: If the standard deviation of the background was used as threshold, shouldn't there be about 16

P26 Tab8: What does "large number of false positives" mean. How large is large?

P27L29: Why should the higher emissions and the not-considering of emission profiles by Pillai et al., result in a larger number of usable plumes? To my knowledge, they analyzed each sufficiently cloud free overpass; please check. Isn't it expected, that you find less usable scenes simply because inverse modeling studies usually assume knowledge of the plume position while you have to search for strong enough signals?

P28L4: ... and also due to the fact that NO2 is used primarily for plume detection but XCO2 will also be used for flux estimation.

P30L7,L8: Which effect dominates, the higher SNR or the lower sensitivity to clouds (see general comment)?

P30L24,L25: ...and cleaner sources in terms of NO2 emissions.

P30L26: Please add a discussion that this future plume detection scheme will have to

solve the problem of the unknown BG before it can be applied to real data.

**3  Technical corrections**

P3L1: can be detected -> is expected to be detected

P3L1: CO2 (or NO2 or CO) -> CO2, NO2, or CO

P7L2: Please add Eq. crossing times to Tab3.

P8L3: "... requirements for Sentinel-5". Please add a reference if possible.

P10 Fig3 Caption: "(a) A large (red) and small (blue) ..." Have you mixed up large and small?

P11L7: "at a satellite" -> "at a smoothed satellite"

P11L12+L13: "n" -> "ns"

P27L15: "which we describe in another publication" do you mean "which we will describe in another publication"

P27L17: "used our study" -> "used in our study"

P27L29: remove "using"

**4  References**

M. Reuter, M. Buchwitz, A. Hilboll, A. Richter, O. Schneising, M. Hilker, J. Heymann, H. Bovensmann, J.P. Burrows: Decreasing emissions of NOx relative to CO2 in East Asia inferred from satellite observations. Nature Geoscience, doi:10.1038/ngeo2257, 2014

---

## Referee Comment (RC2) · Anonymous Referee #2 · 16 Aug 2019

To the authors of the manuscript,

This study assessed the utility of future space instruments to detect CO2 plumes from cities and power plants. The study examined different instruments (CO2, NO2 and CO) with different noise levels under different constellation scenarios for the CO2M. The study is based on a high resolution Nature run achieved by the COSMO model (described in Brunner et al. 2019) that allowed the authors to consider realistic concentration fields and the cloud cover information. A key part of this study is a plume detection algorithm the authors developed. This study examined the plume detection capability of different instruments, assuming the use of the plume detection algorithm.

The use of co-emitted species for assessing anthropogenic CO2 emissions was studied previously in Reuter et al. (2014). But, as pointed out by the referee #1, the SCIAMACHY's large footprint wasn't ideal for observing CO2 concentration enhancements due to localized sources. In my opinion, assessing the utility of current/future improved instruments and/or combinations of these instruments from CO2 monitoring perspective is important. I believe this study has important implications for the planning of future carbon observing missions. These missions will improve our understanding of anthropogenic CO2 emissions.

Before I recommend this manuscript for publication, I would like to raise a few points for discussion, which I listed below. I also included line-by-line comments. I hope my review comments prove helpful to improve this manuscript. I look forward to receiving the authors' response.

Sincerely,

Anonymous referee

Points of discussion:

1. How to quantify the emissions?

Clearly, plume detection is an important first step to quantify emissions from localized sources, but it is not the ultimate goal of satellite carbon missions such as CO2M. In my opinion, it would be helpful for AMT readers if the authors could detail the entire carbon emission quantification process for emission verification support. I am aware that the authors stated that they plan to describe their inverse emission estimation method in another publication. However, I think including some text describing how the detected plume information/data will be used in the inverse estimation is necessary in this manuscript for several reasons. First, as I mentioned earlier, I think it is necessary for the AMT readers to understand the entire picture of the carbon emission estimation. By doing so, the authors should be able to better highlight the significance of this work.

Second, the way carbon emission estimates are obtained (in other words, how the data collected with the plume detection is used) should define good plume detection (which is only loosely defined in this study). I thought the lack of text regarding the entire emission estimation process made it very difficult to define what is useful.

2. What is useful?

Relating to the comment above, the authors should try to define the evaluation criteria more clearly and make it more relevant to carbon emission estimation. In the manuscript, for example, the number of pixels detected was used as a measure. Even if the same number of pixels are detected, the usefulness of the collected information is not necessarily the same in terms of emission estimation.

3. Weak relevance to emission monitoring.

I believe CO2M will be useful for emission monitoring by the planned improvements over previous missions, but I am not quite sure how exactly CO2M will contribute to improving understanding of carbon emissions. This is related to discussion points #1 and #2 that I discussed above. For example, would it be possible for this CO2M (assuming the detection and inverse methods work perfectly) to distinguish underreported power plants?

4. Limitations of the plume detection algorithms?

It seems that the proposed plume detection algorithm requires accurate locations of coal-fired power plants (is this correct?). The source location information is required for the step 4, where the plume is related to the source of interest. This study used the TNO emission inventory that includes reported power plant information for Europe (which I assume is reasonably accurate for this application). But what if this plume detection algorithm is implemented globally? For some parts of the world (where emission uncertainty is thought to be high due to poor data availability), it is often challenging to collect accurate/reliable emission information, such as point source information

and city inventories (this is a part of our motivation to provide an emission verification support, right?). It is thus not easy to build a good inventory with the reported point source emissions like the TNO emission inventory. Can we use this algorithm to detect missing power plants? This is a huge limitation when the monitoring is implemented globally.

5. Overestimation of the success rate due to some simplifications done in this study.

As the authors acknowledged, there are some simplifications made in this study (e.g. aerosols, clouds, emission temporal variation). I understand the necessity of simplicity in the interest of time. However, at the very least, the author should discuss the impact of the simplifications on the results (e.g. successful rate of detection).

Line-by-line comments:

P1, L1: Probably better to add the motivation for detecting plumes? While this may be clear to some AMT readers who are aware of the research concerning space-based CO2 monitoring, it may not be apparent to other readers.

P1, L8: Spell out

P2, L7: We need emission inventories (which are the basic tool for emission monitoring) first before science-based monitoring systems like COM2 (I think). Independent from what? Monitoring system is a little too much for them, although that is something that the scientific community can assist with.

P2, L11: Curious to hear how to directly approach to emissions at the national level. Unlike cities and power plants, it seems to be difficult to do so.

P2, L17: The detection is not the end of the emission estimation problem, but just a start. . .

P4, L11: I assume there are mismatches in surface emission configurations among models (which are probably not important in this study)

P5, P3: Data source for Berlin inventory? How are they constructed? Basic specification?

P8, L14: Adding some info about the CarbonSat mission concept and others would be helpful for the readers who are not familiar with the space-based CO2 monitoring.

P12, L4: So this algorithm requires the source location as a prior information. Correct?

P14, L20: 100 cloud free observations - This might be good enough to cover the source areas spatially, but might not be enough to estimate CO2 emissions.

P27, L12: But this paper still describes what this detectability means in the emission monitoring via inverse modeling. . . the authors at least show the relevance. . .

P27, L22: "useful" - I do think this will be useful, but the authors should try to be a little bit clearer in defining what is useful and why it is useful as this is a scientific paper.

P29, L2- but NO2 and CO2 are not exactly the same although they are co-emitted, right?

P29, L8 – and accurate database of NOx power plants,

P30-L22 huge - this sounds a little bit subjective to me.

P30, L25 weaker – Maybe the authors could be more quantitative?

References

Bovensmann et al. (2010) AMT Nassar et al. (2017) GRL

---

## Author Comment (AC1) · 16 Sep 2019

We would like to thank reviewer #1 for the helpful review and revised the manuscript based on the comments. Please find our detailed response attached and also note the response to reviewer #2.

Please also note the supplement to this comment:
https://www.atmos-meas-tech-discuss.net/amt-2019-180/amt-2019-180-AC1-supplement.pdf

---

## Author Response (AR1)

We would like to thank reviewer #1 for the helpful review and revised the manuscript based on their comments. Please find our detailed response and the revised manuscript below. Please also note the response to reviewer #2.

**General comments**

**1. Reviewer #1:** The proposed method cannot be applied to real satellite data as it is now, because a key problem is not addressed, namely how to derive the background and its uncertainty. This should be discussed more prominently also in the conclusions and already be mentioned in the abstract. To me, many points in the description of the algorithm are unclear (see my specific comments).

**Response:** We revised and restructured the description of our algorithm (Section 2.2.1) making everything hopefully more clear.

In our study, we estimate background and its variance from the corresponding component in the COSMO-GHG simulations. We think that this approach is sufficiently accurate for comparing the detectability of $CO_2$ plumes with $CO_2$, $NO_2$ and CO observations, which was the main objective of this paper. We already discuss the effect of "this optimistic assumption" on the number of detectable plumes in Section 4.3 (Limitations of current plume detection algorithm). We further modified Section 4.3 to clarify our main objective:

>>> *The main objective of this study was to compare the basic detectability of $CO_2$ plumes with $CO_2$, $NO_2$ and CO observations. Our plume detection algorithm was able to detect* weak signals ... <<<

We also replaced "new algorithm" and "novel algorithm" with "simple algorithm" in the manuscript for the same reason. We agree that further development of the plume detection algorithm would be required for its application to real observations. In particular, the background would need to be derived differently. For example, it could be taken from a model or estimated directly from the satellite observations (see Section 2.2.1). We already discuss these options briefly in Section 4.3 and extended the section to provide more details:

>>> *The algorithm assumed accurate knowledge of the mean and variance of the background, which were estimated directly from a simulated background tracer. [...]*

*When applied to real satellite observations the background and its variance could either be taken from a model or estimated directly from the satellite observations. Although atmospheric transport models have large uncertainties at the level of individual plumes, they could provide reasonable estimates of the $CO_2$ and $NO_2$ background. Likely the best option is to derive the background directly from the satellite data, which requires further development of the plume detection algorithm presented here. An improved algorithm could start with an a priori estimate of plume location and background and would then be updated iteratively to improve both plume location and background. A model could be used here to determine a suitable a priori plume location and background. An improved version of the algorithm presented in this paper has the potential for increasing the number of detectable plumes per satellite as well as the number of $CO_2$ pixels per plume.* <<<

**2. Reviewer #1**: NO2 can be used in more cloud contaminated scenes compared to CO2. This is one reason why you find more plumes in NO2. When computing the number of successfully found inner-plume soundings with NO2, do you apply the stricter CO2 cloud filter in the last step? This would be very important, because only these soundings could be used for the flux estimation. Please clarify this in the manuscript and if this last filter is not applied, discuss why you think that your results are still meaningful for CO2M.

**Response**: Yes, we do apply the stricter CO2 cloud filter in the last step. The TP and FP values are thus only computed for cloud-free CO2 pixels. We added the following sentence to Section 2.2.2 (Performance evaluation):

>>> *To remove the impact of different cloud thresholds, TP, FP and PPV were only computed for cloud-free pixels using a cloud threshold of 1%, because we are primarily interested in valid CO2 observations that can be used for estimating CO2 emissions."* <<<

**3. Reviewer #1**: Which effect is dominating the advantages of NO2, less strict cloud filtering or better SNR? How would both compare if CO2 cloud filtering would have been also applied to NO2?

**Response**: Since we did not apply the plume detection to NO2 observations with a 1% cloud threshold, we cannot answer the question quantitatively. We expect that even with a 1% cloud threshold the NO2-based plume detection would still perform significantly better because of the higher signal-to-noise ratio (see Table 6).

**4. Reviewer #1**: In respect to a potential future application to real data, the pros and cons of plume detection based flux retrieval methods should be compared to inverse modeling approaches in more depth. Some examples: Inverse modeling can be applied also to plumes below the detection limit. Inverse modelling may place the plumes wrongly so that modelled plumes become less correlated with measured plumes resulting in underestimated fluxes. Plume detection depends on plume strength, so that this may introduce an observational bias towards large fluxes, i.e., annual average fluxes will likely be biased high, so that comparisons of reported and measured fluxes are more meaningful on a per-overflight basis. Plume detection methods can be applied to unknown source positions. Etc.

**Response**: We would like to thank the reviewer for these insightful points. We included some of the more general points in the introduction and also updates Section 4.3 adding some more specific points:

>>> *The system would allow for observing CO2 plumes of individual point sources such as large cities and power plants and for quantifying the respective emissions during single satellite overpasses (Bovensmann et al. 2010; Pillai et al. 2016, Velazco et al. 2011). A CO2 plume is defined here as an enhancement of CO2 concentrations above the background in the satellite image caused by the emissions of a given source. The emissions of the source can be estimated from the CO2 enhancement inside the plume, which requires that the plume location is identified in the satellite observations. An atmospheric transport model may be used for simulating the plume location and for estimating the emissions with an inversion framework (e.g. Pillai et al. 2016; Broquet et al. 2018). However, the simulated plume might be*

*significantly displaced due to uncertainties in wind fields and emission heights, which would result in systematic errors in the estimated emissions (Broquet et al. 2018, Brunner et al. 2019). It is therefore desirable to detect the plume directly in the satellite observations, which would make it possible to correct transport-related errors in the simulations but also to estimate the emissions directly from the CO2 enhancements in the plume using plume fitting or mass balance approaches, which only require an estimate of the mean wind speed within the plume {Fioletov et al. 2015; Krings et al. 2013; Varon et al 2018}. While the potential for detecting and estimating emissions from CO2 fluxes has been demonstrated for strong CO2 plumes of megacities and large point sources using* the Orbiting Carbon Observatory 2 (OCO-2, Crisp et al. 2017) (Nassar et al 2017, Reuter et al 2019), a major challenge is... <<<

**5. Reviewer #1**: Why do you consider PPV a good measure for performance? As an example: consider a very conservative plume detection algorithm, detecting only soundings which are 20ppm larger than the BG. In this case, TP would admittedly become small, but FP would become zero, so that PPV would become 1. Could the Hanssen-Kuiper Skill Score or True Skill Statistic be a better option?

**Response**: The PPV was used primarily to identify detected plumes with a large number of pixels that were incorrectly assigned to the source. To measure the overall performance, the PPV needs to be used complementary with the TP. A "good" detection would thus need to have both high TP and PPV. The detection in the example has a high PPV but also a low TP.

We did not use the Hanssen-Kuiper Skill Score / True Skill Statistics, because it depends on the number of true negatives (TN), i.e. all pixels outside the true plume. TN cannot be defined unambiguously in our application, because it depends on swath width and the size of the model domain.

Please note that in the paper we do not evaluate the performance of the plume detection algorithm but the performance of the different instrument scenarios for detecting the CO2 plume. We have revised Section 2.2.2 ("Performance evaluation") to better explain this purpose (see also our response to Comment 16).

**6. Reviewer #1**: To my knowledge, the Z-test is appropriate only for samples with more than 30 elements. 5 out of 8 tested smoothing kernels (Fig.4) consist of less elements. Why do you think the z-test is still appropriate in these cases?

**Response**: The Z-test can also be applied to smaller samples, even to single observations, when the variance of the population is known. The sample size is only a problem if the variance has to be estimated from the sample, which is not the case for our application.

**7. Reviewer #1**: What happens at cloud edges and at the edges of the swath? Effectively, the smoothing kernel here has fewer elements. Is this considered?

**Response**: Yes, the n in Eq. (1) is smaller if satellite pixels are missing in the neighborhood. We modified and added the following sentences to explain the difference between n and ns: >>> *The random component would reduce with the inverse square root of the number of valid pixels n in the neighborhood, whereas the systematic error would be approximately independent of n. The number of valid pixels*

*n can be smaller than the size of the neighborhood $n_s$ when pixels are missing, e.g. due to clouds or at the boundary of the satellite swath."* <<<

**Specific comments:**

**8. Reviewer #1**: P2L23: Please also cite Reuter et al. 2014 in this context. They analyzed CO2+NO2 measurements from the same sensor (SCIAMACHY) and proposed multi-species measurements for future satellite missions. As signals in SCIAMACHY data are smaller because of the large pixel size, they had to follow a statistical approach. Note that this does not only cover strong localized sources, but all localized sources.

**Response**: We added the reference.

**9. Reviewer #1**: P3L4-L5: I would suggest moving this outlook to the conclusions section.

**Response**: We think it is better to keep this here to describe the context within the SMARTCARB project.

**10. Reviewer #1:** P3L8-L27: Strictly speaking, everything in these lines does neither belong to "data" nor to methods. Maybe it would fit better in the introduction.

**Response**: We moved the section to the introduction.

**11. Reviewer #1**: P3L24: The situation is also not that simple for individual point sources, because chemistry changes the NO2 plume which is not the case for the CO2 plume. The following points should be addressed in this context: 1) NO2 decays with time. 2) NO2 and CO2 can be emitted/transported into various altitudes with differences chemical regimes. For example, the high altitude part of the plume may decay faster/slower compared with the low altitude part of the plume. As the wind direction often changes with height, the satellite observed CO2 plume may have a different (wider) shape as the NO2 plume. 3) NO2 is primarily emitted as NO (which later forms NO2), which can result in different plume shapes especially near the source.

**Response**: Our simulations account for (1) the NO2 decay with time with a constant exponential decay and (2) the conversion of NO to NO2 by emitting and transporting NO2 as part of a NOx tracer. The last paragraph of the introduction has been changed as follows:

>>> *However, this requires that the plumes of CO2 and of the auxiliary trace gas are spatially congruent. This is usually the case when they are emitted from the same source, for example, a power plant. The shape of the NO2 plumes might deviate from the CO2 plume for two reasons. First, NO2 is emitted mainly as nitrogen monoxide (NO), which is converted to NO2 over time resulting in lower NO2 concentrations near the source. Second, NO2 decays slowly with time reducing NO2 concentrations downstream. To account for these two effects, we simulated nitrogen oxides (NOX = NO + NO2) that slowly decays with time and calculated NO2 from NOX concentrations offline by applying a formula frequently used to represent NO2:NOX ratios downstream of emission sources (Düring et al. 2011). The situation is more complex for cities where the emissions originate from different sectors (industry, heating, transport*

*etc.) that emit with different temporal profiles and at different altitude levels, and which have different emission ratios (Brunner et al. 2019). In this study, we therefore carefully consider the vertical and temporal profiles of emissions from different sectors, which makes it possible to test for congruence.* <<<

*We also added the following sentence to Section 2.1.1*

>>> *NO2 concentrations were calculated from NOX concentrations using an empirical formula used frequently for representing NO2:NOx ratios downstream of emission sources (Düring et al. 2011).* <<<

and the following sentence to the abstract:

>>> CO2 and NO2 plumes were found to overlap to a large extent, although NOX had a limited lifetime (assumed to be 4 hours) and although CO2 and NOX were emitted with different NOX:CO2 emission ratios by different source types with different temporal and vertical emission profiles. <<<

The effect of different chemical regimes on the plumes shapes might be important, but we are not aware of any publication studying the effect or showing that it has a significant effect on the congruence of CO2 and NO2 plumes. Our study is also not able to quantify this effect, because it requires full chemistry transport simulations, which is computational too expensive for one year of simulation. Because we think that the introduction is not a good place to discuss this, we added the following sentence to the discussions (Section 4.2) instead:

>>> *Our study could not simulate detailed NOx chemistry due to the high computational costs. We can therefore not rule out a larger mismatch between some CO2 and NO2 plumes than simulated here, for example, because of different NO2 decay rates at different altitudes. However, Reuter et al. (2019) found no obvious mismatch when comparing co-located CO2 and NO2 plumes from OCO-2 and TROPOMI observations. A follow-up study for quantifying the effect of NOx chemistry would be certainly desirable.* <<<

**12. Reviewer #1**: P7L4: sun-synchronous orbits are not circular.

**13. Reviewer #1**: P7L5: "... in case of 3 satellites." Please describe what this means (the satellites will have the same Eq. crossing times...).

**Response**: We re-formulated the paragraph for clarification:

>>> The basic assumption for a constellation is that the individual satellites are spaced with equal angular distance in the same orbit with the same orbit parameters, for instance separated by 180°, 120° and 90° on a full circle in case of 2, 3 and 4 satellites. <<<

**14. Reviewer #1**: P9L30: Cloud coverage is a main driver for data yield of XCO2 satellite products. I guess COSMO diagnoses fractional cloud coverage in its 3D model domain. How do you/COSMO compute the 2D cloud fractional coverage from this (e.g., maximum/minimum overlap assumption) and how realistic are the 2D cloud coverage statistics for this particular model?

**Response**: We'd like to thank reviewer #1 for this important comment. Cloud cover is computed by COSMO assuming minimum overlap. We investigated how cloud cover in COSMO compares with MODIS satellite data and added the following sentences to the method section proving more details on COSMO cloud fraction and the reason why it was used in our study:

*>>> Previous studies used the MODIS cloud mask product available at 1 km resolution (Ackermann et al. 2007) for masking cloudy CO2 observations (Buchwitz et al. 2013, Pillai et al. 2016). Since CO and NO2 observations can tolerate larger cloud fractions, a cloud fraction product would be needed for masking pixels with different thresholds but the MODIS cloud product is only available at 5 km resolution (Platnick et al. 2007). Therefore, we used total cloud fractions computed by COSMO-GHG, i.e. the same model as used for the tracer transport simulations, that are available at model resolution. COSMO-GHG computes total cloud fraction from layer cloud fractions assuming minimum overlap. The differences between clouds masks derived from COSMO-GHG and MODIS products and their effects on data yield are discussed in Section 4.1. <<<*

Furthermore, we added a paragraph to Section 4.1 comparing the difference in data yield using COSMO cloud fractions and the MODIS cloud mask product.

*>>> Clouds have a strong impact on the number of cloud-free observations. A major difference between Pillai et al. (2016) and our study are the different approaches for masking cloudy observations. Pillai et al. (2016) used the MODIS cloud mask product (MOD35_L2) while we used cloud masks derived from cloud fractions simulated with the COSMO-GHG model. To compare these two approaches, we computed the number of cloud-free pixels using the MODIS cloud mask product (MOD35_L2), the MODIS cloud product (MOD06_L2) and COSMO-GHG simulations. The COSMO-GHG cloud fractions were spatially averaged over the MODIS pixels. The comparison shows that monthly fractions of cloud-free pixels agree well between masks computed from COSMO-GHG and MODIS cloud fractions for both a 1% and 30% threshold, while the cloud-free pixels based on the MODIS cloud mask product are about twice as high (Figure S1). The larger fraction of cloud-free observations obtained from the MODIS cloud mask product is also consistent with a validation study showing that the product is not very sensitive to optically thin clouds (Ackermann et al. 2008). The differences likely explain the different number of potentially useful orbits between Pillai et al 2016 and our study. It also suggests that Pillai et al (2016) overestimate the number of potentially useful orbits while our results are likely more accurate. <<<*

We will add the following figures to the supplement of the paper:

[Figure]

*Figure S1: Monthly fractions of cloud-free pixels using COSMO- and MODIS-based cloud masks with different spatial resolution (1 km and 5 km at nadir) and (a) 1% and (b) 30% cloud fraction as threshold.*

The conclusions were also updated to include the new result:

*"This number is somewhat smaller than reported in earlier studies (Bacour et al. 2015, Pillai et al., 2016), mainly because masking cloudy pixels based on the simulated cloud fields leads to less cloud-free observations than using the MODIS cloud mask product."*

**15. Reviewer #1**: P10 Fig3 Caption: "... are marked with "T" for True." Shouldn't be True = blue/red and

T = detected by Z-test? If so, consider replacing T by Z.

**Response**: "T" was used to mark pixels detected by the Z-test. We changed "T" to "D" to avoid confusion with the true plume and changed caption and legend accordingly.

**16. Reviewer #1**: P11L7: The choice of neighborhood size appears a bit arbitrary. Please discuss, why it makes sense to try neighborhood sizes as selected. For example, I could imagine, that the maximum meaningful neighborhood size is reached once the noise of the smoothed pixels dropped significantly below levels of typical enhancements. On the other hand, the neighborhood should probably never be significantly larger than the typical plume extends.

**Response**: The performance of different neighborhoods was analyzed in more detail in the SMART-CARB final report (see https://www.empa.ch/documents/56101/617885/FR_Smartcarb_final_Jan2019.pdf), which is also cited in the manuscript. We added the following sentences to the 2nd paragraph of Section 3.2 ("Signal-to-noise ratios") to provide some more details:

>>> A large neighborhood reduces the random noise of the measurements and therefore allows detecting smaller signals. On the other hand, a large neighborhood will include background values in the computation of the averages at the plume edges and reduce the signal. The sizes used here roughly correspond to the typical diameters of the CO2 plumes from Berlin (about 15 km) and the power stations (about 6 km), respectively, and were also found most suitable for the plume detection algorithm, because they maximize TPs without reducing PPVs too much (see also Kuhlmann et al. 2019). <<<

**17. Reviewer #1**: P11L20,L21: Please describe why the Z-test calculates the SNR (e.g., the variance of the basic population is assumed to be known from the measurement errors, Xobs is an estimator for the expectation value).

**Response**: We added a more detailed description of the z-test and it is relation to the signal-to-noise ratio (see revised manuscript).

**18. Reviewer #1**: P11Eq1: On P9L4 you stated that systematic errors are not scope of this study. Why considering them in Eq1?

**Response**:  The equation (Eq. 2 in the revised manuscript) provides a general form that considers both random and systematic errors. In our study, we do not consider systematic errors in the satellite observations but we still consider systematic errors in the background field resulting from the biosphere or other anthropogenic sources.

**19. Reviewer #1:** Xobs are the smoothed observations (not to be confused with the original observations). This should be corrected also at P11L13.

**Response**: We changed this to "spatially averaged satellite observations"

**20. Reviewer #1:** Sigma_rand is only approximately the same for all soundings within the smoothing kernel but strictly speaking; it is different for each sounding.

**Response**: This is correct. We used a constant value, because the random error in real data is also not known exactly.

**21. Reviewer #1**: P11Eq2: Does this imply that there will also be false positives just by chance? How often is this expected to happen?

**Response**: Yes, since we used $q=0.99$, we expect that 1% of detected pixels are false positives. However, the number of false positives is reduced by labelling in the second step of the algorithm.

**22. Reviewer #1**: P11L30: Does this mean $sigma_{sys}$ = $sigma_{bg}$ in Eq 1?

**Response**: Yes, under the assumption made in our study.

**23. Reviewer #1**: P12L1: What is the chosen level for significance?

**Response**: In this study, we used $q=0.99$.

**24. Reviewer #1**: P12L5: Please explain "Moor neighborhood".

**Response**: In (digital) image processing, a "Moore neighborhood" is a neighborhood where each pixel has eight potential neighbors (see for example https://en.wikipedia.org/wiki/Pixel_connectivity). We restructured the paragraph slightly to improve readability:

*>>> In the second step, pixels that are enhanced (``true'') and connected are assumed to belong to the same plume. We label regions of connected pixels using a standard labeling algorithm. Neighboring pixels are identified using a Moore neighborhood, where each pixel has eight potential neighbors. Each region is assigned a unique integer value. <<<*

**25. Reviewer #1**: P12L9: "… radius of 5km" In reality, this can result in non-detected NO2 plumes when NO2 is emitted primarily as NO.

**Response:** Yes this is correct and we accounted for this by calculating NO2 columns from NOx fields with an empirical formula (see our reply to comment #11). It should be noted that we did not find a scene where a radius of 5 km was not sufficient for assigning a detected (cloud-free) plume to the source.

**26. Reviewer #1**: P12L21: Can you separate both effects; which one usually dominates? In other words, why is a plume not detected, too low signal or CO2 plume != NO2 plume?

**Response**: The NO2-based plume detection often fails, if the plume overlaps with neighboring plumes which results in low PPV or if the scene is extremely cloudy. The NO2 signal-to-noise ratio is generally high and the CO2 and NO2 plume match well in the simulations.

**27. Reviewer #1**: P13Fig.5d: How can you compute Xobs in totally cloudy regimes?

**Response**: $X_{obs}$ cannot be computed for fully cloudy scenes. Figure 5 shows examples of true CO2 plumes taken from the synthetic observations. We modified the caption for clarification.

**28. Reviewer #1**: P14L14: If the standard deviation of the background was used as threshold, shouldn't there be about 16

**Response**: Sorry, but the question seems incomplete.

**29. Reviewer #1**: P26 Tab8: What does "large number of false positives" mean. How large is large?

**Response**: A computation of the number of false positives was not possible due to the missing true plume for Janschwalde. Instead, visual inspection was used, but "large number of false positives" looks typically like Figure 13c. We now cite Fig. 13c as example both in the caption as well as in the text.

**30. Reviewer**: P27L29: Why should the higher emissions and the not-considering of emission profiles by Pillai et al., result in a larger number of usable plumes? To my knowledge, they analyzed each sufficiently cloud free overpass; please check. Isn't it expected, that you find less usable scenes simply because inverse modeling studies usually assume knowledge of the plume position while you have to search for strong enough signals?

**Response**: In this section, we compare potentially observable CO2 plumes independent of the detection algorithm and therefore expected that our numbers would be similar to potentially useful plumes identified by Pillai et al. (2016). After checking Pillai et al (2016), we decided that it might be more reasonable to compare their number of potentially useful orbits (41 for 500-km swath) instead of the number of orbits with successful emission estimates (27 for 500-km swath). We therefore revised the paragraph as follows:

>>> *For Berlin, we estimated that 3 to 13 potentially useful CO2 plumes (defined as plumes with at least 100 cloud-free pixels above a threshold of 0.05 ppm) would be observable, but not necessarily detectable, during one year by a single CO2M satellite with a 250-km wide swath. Pillai et al. (2016) identified 41 potentially useful orbits for estimating emissions with a 500-km wide swath. Although a direct comparison of these two numbers is difficult because of the different swath widths and the different definitions of "usefulness", we can still conclude that our study identified significantly fewer plumes than Pillai et al. (2016) even after halving their number to account for their wider swath.* <<<

Next, we add new paragraph showing that the difference can be mostly be explained by differences in the cloud masks used in our and their study (see response to Comment 14).

Higher emissions and not-considering emission profiles result in more potentially useful plumes, because they result in larger plumes that are less likely fully covered in clouds. We revised the paragraph as follows:

>>> *Further differences are to be expected because small plumes with less than 100 pixels were excluded in our case and because of different meteorology, especially cloud cover and wind speed, for the different simulation periods. In addition, Pillai et al. (2016) used higher emissions and did not consider vertical profiles of emissions, which results in stronger and correspondingly larger plumes, which are less likely to be fully covered by clouds (Brunner et al., 2019).* <<<

**31. Reviewer #1**: P28L4: ...and also due to the fact that NO2 is used primarily for plume detection but XCO2 will also be used for flux estimation.

**Response**: Sorry, we are not able to relate the comment with our manuscript.

**32. Reviewer #1**: P30L7,L8: Which effect dominates, the higher SNR or the lower sensitivity to clouds (see general comment)?

**Response**: (see our reply to Comment 3)

**33. Reviewer #1**: P30L24,L25: ... and cleaner sources in terms of NO2 emissions.

**Response**: The sentence now reads: "Therefore, adding an NO2 instrument is highly recommended and the low-noise instrument is preferable for detecting also weaker and cleaner plumes in terms of NO2 emissions."

**34. Reviewer #1**: P30L26: Please add a discussion that this future plume detection scheme will have to solve the problem of the unknown BG before it can be applied to real data.

**Response**: (see our reply to Comment 1)

**Technical corrections**

**Reviewer #1**: P3L1: can be detected -> is expected to be detected

**Response**: Done

**Reviewer #1**: P3L1: CO2 (or NO2 or CO) -> CO2, NO2, or CO

**Response**: Done.

**Reviewer #1**: P7L2: Please add Eq. crossing times to Tab3.

**Response:** Equator crossing times are already in Table 2.

**Reviewer #1**: P8L3: "...requirements for Sentinel-5". Please add a reference if possible.

**Response**: We added the following reference:

Ingmann et al (2012): Requirements for the GMES Atmosphere Service and ESA's implementation concept: Sentinels-4/-5 and -5p, doi: 10.1016/j.rse.2012.01.023.

**Reviewer #1**: P10 Fig3 Caption: "(a) A large (red) and small (blue) …" Have you mixed up large and small?

**Response**: Yes, the caption is correct now.

**Reviewer #1**: P11L7: "at a satellite" -> "at a smoothed satellite"

**Response**: Changed to: "at a spatially smoothed satellite pixel."

**Reviewer #1**: P11L12+L13: "n" -> "ns"

**Response**: We modified the sentence (see our previous answer).

**Reviewer #1**: P27L15: "which we describe in another publication" do you mean "which we will describe in another publication"

**Response**:  Changed.

**Reviewer #1**: P27L17: "used our study" -> "used in our study"

**Response**: Changed.

**Reviewer #1**: P27L29: remove "using"

**Response**: Removed

We would like to thank Reviewer #2 for their helpful comments. We revised our manuscript based on his/her suggestions and provide a point-to-point response below. Please also note the response to Reviewer #1 and the revised manuscript showing the changes in the manuscript below.

**General comments**

**1. Reviewer #2**:

1. How to quantify the emissions?

Clearly, plume detection is an important first step to quantify emissions from localized sources, but it is not the ultimate goal of satellite carbon missions such as CO2M. In my opinion, it would be helpful for AMT readers if the authors could detail the entire carbon emission quantification process for emission verification support. I am aware that the authors stated that they plan to describe their inverse emission estimation method in another publication. However, I think including some text describing how the detected plume information/data will be used in the inverse estimation is necessary in this manuscript for several reasons. First, as I mentioned earlier, I think it is necessary for the AMT readers to understand the entire picture of the carbon emission estimation. By doing so, the authors should be able to better highlight the significance of this work. Second, the way carbon emission estimates are obtained (in other words, how the data collected with the plume detection is used) should define good plume detection (which is only loosely defined in this study). I thought the lack of text regarding the entire emission estimation process made it very difficult to define what is useful.

**Response**: We extended the introduction to provide readers with more details on CO2 emission quantification methods for cities and point sources:

>>> *The system would allow for observing CO2 plumes of individual point sources such as large cities and power plants and for quantifying the respective emissions during single satellite overpasses (Bovensmann et al. 2010; Pillai et al. 2016, Velazco et al. 2011).* *A CO2 plume is defined here as an enhancement of CO2 concentrations above the background in the satellite image caused by the emissions of a given source. The emissions of the source can be estimated from the CO2 enhancement inside the plume, which requires that the plume location is identified in the satellite observations. An atmospheric transport model may be used for simulating the plume location and for estimating the emissions with an inversion framework (e.g. Pillai et al. 2016; Broquet et al. 2018). However, the simulated plume might be significantly displaced due to uncertainties in wind fields and emission heights, which would result in systematic errors in the estimated emissions (Broquet et al. 2018, Brunner et al. 2019). It is therefore desirable to detect the plume directly in the satellite observations, which would make it possible to correct transport-related errors in the simulations but also to estimate the emissions directly from the CO2 enhancements in the plume using plume fitting or mass balance approaches, which only require an estimate of the mean wind speed within the plume {Fioletov et al. 2015; Krings et al. 2013; Varon et al 2018}. While the potential for detecting and estimating emissions from CO2 fluxes has been demonstrated for strong CO2 plumes of megacities and large point sources using* *the Orbiting Carbon Observatory 2 (OCO-2, Crisp et al. 2017) (Nassar et al 2017, Reuter et al 2019), a major challenge is...* <<<

**2. Reviewer #2**:

2. What is useful?

Relating to the comment above, the authors should try to define the evaluation criteria more clearly and make it more relevant to carbon emission estimation. In the manuscript, for example, the number of pixels detected was used as a measure. Even if the same number of pixels are detected, the usefulness of the collected information is not necessarily the same in terms of emission estimation.

**Response:** The number of detectable pixels was used as primary evaluation criteria, because it makes it possible to compare the performance of CO2- and NO2-based plume detection for a given source without including further parameters that would affect the estimation of CO2 emissions like source strength and meteorology. These effects could only be accounted for when emissions were actually estimated from the detected plumes, which is outside of the scope of our paper. We added the following sentences to Section 3.1 ("Coverage and potential for plume detection") to provide the reader with the reasons for using the number of detected pixels:

>>> *To cover at least the whole city area, we only consider CO2 plumes with at least 100 cloud-free CO2 pixels to be useful. [...]. It should be noted that this number of pixels is not necessarily sufficient for estimating the emissions of a source with certain accuracy, which depends, among others, on instrument precision, meteorology and source strength. Nonetheless, detecting the full crosswind diameter is the minimum requirement, for example, for flux-based inversion methods (e.g. Krings et al., 2013; Reuter et al., 2019). The number of detected pixels is a useful measure for comparing the detectability of CO2 plumes with CO2, CO and NO2 observations, because source strength and meteorology are the same for a given source.* <<<

**3. Reviewer #2**:

3. Weak relevance to emission monitoring.

I believe CO2M will be useful for emission monitoring by the planned improvements over previous missions, but I am not quite sure how exactly CO2M will contribute to improving understanding of carbon emissions. This is related to discussion points #1 and #2 that I discussed above. For example, would it be possible for this CO2M (assuming the detection and inverse methods work perfectly) to distinguish underreported power plants?

**Response**: We think it is outside the scope of this paper to explain how CO2M can contribute to emission monitoring, but instead would keep the paper focused on the detectability of CO2 emission plumes using CO2, NO2 and CO observations. These questions are the focus of various currently ongoing studies like CHE (www.che-project.eu) and verify (https://verify.lsce.ipsl.fr/). Many details on the scope of CO2M were also provided in the reports by Ciais et al. (2015) and Pinty et al. (2018) cited in our paper.

**4. Reviewer #2**:

4. Limitations of the plume detection algorithms?

It seems that the proposed plume detection algorithm requires accurate locations of coal-fired power plants (is this correct?). The source location information is required for the step 4, where the plume is related to the source of interest. This study used the TNO emission inventory that includes reported power plant information for Europe (which I assume is reasonably accurate for this application). But what if this plume detection algorithm is implemented globally? For some parts of the world (where emission uncertainty is thought to be high due to poor data availability), it is often challenging to collect accurate/reliable emission information, such as point source information

**Response**: The location of the source is only required for assigning a plume to a certain source. If no accurate source location is available, it is still possible to detect regions of enhanced $CO_2$ or $NO_2$ observations with our algorithm. For unknown sources, it should be possible to identify their exact location, for example, by looking at aerial or satellite images, which are available for most populated regions (e.g. GoogleEarth). Alternatively, unknown sources could also be detected in long-term averaged $CO_2$ or $NO_2$ satellite images.

**5. Reviewer #2:**

5. Overestimation of the success rate due to some simplifications done in this study.

As the authors acknowledged, there are some simplifications made in this study (e.g. aerosols, clouds, emission temporal variation). I understand the necessity of simplicity in the interest of time. However, at the very least, the author should discuss the impact of the simplifications on the results (e.g. successful rate of detection).

**Response**: The potential influences of the simplifications on the results are discussed in Section 4 ("Discussion"), which includes a discussion on the impact on the successful detections, for example, due systematic errors. We updated Section 4.2 accordingly:

>>> *Our study did not include systematic errors in the satellite observations from aerosols, clouds and surface reflectance, which can result in spatial patterns resembling plume structures and therefore complicate plume detection. We therefore might overestimate the number of detectable plumes. Although systematic errors affect all satellite products, the effect would be more severe for XCO2 than for NO2 due to the much smaller signal-to-noise ratios. How much aerosols enhance measurement uncertainties and correspondingly reduce the ability to detect plumes cannot be quantified here, but will be studied, for example, in a study on the use of aerosol information for estimating fossil fuel CO2 emissions (AEROCARB) performed by a consortium led by SRON.* <<<

We also added a new paragraph discussing the impact of clouds on the number potentially observable plumes (see our reply to Comment 14 of reviewer #1) and a paragraph on possible effects of NOX chemistry (see our reply to Comment 11 of reviewer #1).

**Line-by-line comments:**

**Reviewer #2:** P1, L1: Probably better to add the motivation for detecting plumes? While this may be clear to some AMT readers who are aware of the research concerning space-based CO2 monitoring, it may not be apparent to other readers.

**Response**: We updated the introduction to provide a more detailed motivation for plume detection (see our reply to Comment 1).

**Reviewer #2:** P1, L8: Spell out

**Response**: Changed to "were simulated [...] with the Consortium for Small-scale Modeling - Greenhouse Gases (COSMO-GHG) model"

**Reviewer #2:** P2, L7: We need emission inventories (which are the basic tool for emission monitoring) first before science-based monitoring systems like COM2 (I think). Independent from what? Monitoring system is a little too much for them, although that is something that the scientific community can assist with.

**Response**: We changed the sentence as follows: "However, many cities are currently lacking a CO2 emission monitoring system to evaluate their policies."

**Reviewer #2:** P2, L11: Curious to hear how to directly approach to emissions at the national level. Unlike cities and power plants, it seems to be difficult to do so.

**Response**: Since countries typically do not have emission plumes, a different approach is required that does not estimate emission from detected plumes. See for example "Report for mission selection: CarbonSat" p164ff ([https://esamultimedia.esa.int/docs/EarthObservation/SP1330-1_CarbonSat.pdf](https://esamultimedia.esa.int/docs/EarthObservation/SP1330-1_CarbonSat.pdf)).

**Reviewer #2:** P2, L17: The detection is not the end of the emission estimation problem, but just a start ...

**Response**: We provided more information in the introduction (see our response to Comment 1).

**Reviewer #2:** P4, L11: I assume there are mismatches in surface emission configurations among models (which are probably not important in this study)

**Response**: Yes, emissions can be treated quite differently in different model setups. Our emission setup is described in more details by Brunner et al. ([https://doi.org/10.5194/acp-19-4541-2019](https://doi.org/10.5194/acp-19-4541-2019)).

**Reviewer #2:** P5, L3: Data source for Berlin inventory? How are they constructed? Basic specification?

**Response**: We added the reference for Berlin inventory, which is (unfortunately) only available in German)
([https://www.berlin.de/senuvk/umwelt/luftqualitaet/de/emissionen/download/Endbericht_Emissionkataster_2015.pdf](https://www.berlin.de/senuvk/umwelt/luftqualitaet/de/emissionen/download/Endbericht_Emissionkataster_2015.pdf)).

**Reviewer #2:** P8, L14: Adding some info about the CarbonSat mission concept and others would be helpful for the readers who are not familiar with the space-based CO2 monitoring.

**Response**: We added the following sentence to explain Carbonsat:

>>> *Carbonsat was a CO2 imaging spectrometer proposed for ESA's eighth Earth Explorer mission with specifications similar to those of the CO2M mission.* <<<

**Reviewer #2:** P12, L4: So this algorithm requires the source location as a prior information. Correct?

**Response**: (see are reply to the general comments)

**Reviewer #2:** P14, L20: 100 cloud free observations - This might be good enough to cover the source areas spatially, but might not be enough to estimate CO2 emissions.

**Response:** (see our reply to comment 4)

**Reviewer #2:** P27, L12: But this paper still describes what this detectability means in the emission monitoring via inverse modeling... the authors at least show the relevance...

**Response**: We hope that we correctly assume that "this paper" refers to Pillai et al. (2016). Pillai et al. (2016) focus on inverse modelling assuming that the location of the plume is perfectly known from the simulations. They acknowledge this limitation and write that "transport-related errors are [...] non-negligible and should be [...] addressed [...]." Our study focus on the detectability of plumes also because it is an approach to account for transport-related errors in atmospheric simulations. We updated our introduction accordingly (see also our reply to comment 1).

**Reviewer #2:** P27, L22: "useful" - I do think this will be useful, but the authors should try to be a little bit clearer in defining what is useful and why it is useful as this is a scientific paper.

**Response**: (see our reply to the general comments)

**Reviewer #2:** P29, L2- but NO2 and CO2 are not exactly the same although they are co-emitted, right?

**Response**: We added few sentences on the congruence of NO2 and CO2 plumes at the end of the introduction and in Section 4.2.

**Reviewer #2:** P29, L8 – and accurate database of NOx power plants,

**Response**: (see our reply to Comment 2)

**Reviewer #2:** P30-L22 huge - this sounds a little bit subjective to me.

**Response**: We think that the use of "huge" is acceptable here.

**Reviewer #2:** P30, L25 weaker – Maybe the authors could be more quantitative?

[revised manuscript text omitted]

---

## Author Response (AR2)

We like to thank Referee #2 again for reviewing the manuscript, which has been very useful for improving the manuscript. We have modified the manuscript based on his/her comments. Please find below our point-by-point replies as well as the revised manuscript showing the differences.

**General comments:**

**Reviewer #2:**
General comments 3: This study is addressing a component (plume detection) of the CO2M based CO2 monitoring system (= measurement technique). It is important to show this studys relevance and significance in the large picture of CO2 monitoring, especially because the authors have highlighted their relevance to CO2 monitoring. Thus, I still think the authors need to provide how CO2M will be contributing to CO2 monitoring to show this studys significance.

**Authors reply:**
We still consider it outside the scope of this study to provide details on the role of the CO2M mission for quantifying $CO_2$ emissions besides what has been written already in the manuscript. To better convey the objective and significance of our study, we have rewritten the 4th paragraph of the introduction (P3, L11):

> Since the detection of the $CO_2$ plume is a first and important step of a $CO_2$ emission monitoring system, the aim of this paper is to investigate whether and how often $CO_2$ plumes are expected to be detected in the satellite images during a year depending on the size of the CO2M satellite constellation and on instrument error scenarios. The detectability is studied for satellite images of $CO_2$ alone or in combination with images of $NO_2$ and CO to investigate the added value of additional measurements either on the same CO2M satellite ($2x2$ km$^2$, overpass: 11:30 local time) or with the Sentinel-5 instrument ($7.5x7.5$ km$^2$, overpass: 9:30 local time). In this paper, we analyse the signal-to-noise ratios of a city plume and of different point sources for the different instruments. Furthermore, based on a newly developed simple plume detection algorithm, we identify statistically significant plume signals against instrument noise and background variability. The results are used to provide recommendations for the dimensioning of the CO2M mission, which will be a key component of the Copernicus $CO_2$ emission monitoring and verification support system.

**Reviewer #2:**
As CO2M has been a big focus in this study, introducing the satellite with a few sentences by citing other studies does not seem to be sufficient. This is an OSSE study, but we need to understand which instrument/measurement technique the authors studied.

**Authors reply:**
We have re-written Section 2.1.2 ("Satellite instrument scenarios") to introduce the satellite mission in some more detail:

> The CO2M mission is a proposed constellation of satellites flying in a sun-synchronous low-earth orbit with equator crossing times around 11:30 local time. Each satellite will carry an imaging spectrometer measuring in the near-infrared (NIR) and in two short-wave infrared spectral bands (SWIR-1 and SWIR-2) for retrieving $CO_2$ as the main payload. The NIR band is used to retrieve information on the dry air column, on surface pressure and on aerosols and clouds. The SWIR-1 and SWIR-2 bands contain weak and strong absorption features of $CO_2$ and provide additional information on aerosols and clouds, especially on thin cirrus clouds. A $CO_2$ retrieval using these three bands is described for example by ODell et al. (2012). CO2M is planned to carry also additional instruments for measuring $NO_2$, aerosols and clouds. In an

earlier phase, also an instrument measuring CO was considered. The preliminary system concept envisages a pixel size of 4 km$^2$ and swath width of 250 km or more.

**Reviewer #2:**
General comments 5: I am not here questioning the feasibility of detection, but the step you attribute a plume to a certain source. I thought emission attribution is an important component of the CO2 monitoring (thats why I keep asking a question like above...). Yes, what the authors mention is fair and probably theoretically possible. However, I dont think it is practical a nd the authors havent show the ability to maintaining policy-relevant quality. The authors should check how the Google Earth image look like over China... the long term average can be used, but detection is not the last step... anyway.

**Authors reply:**
We think it is outside the scope of this study to show policy-relevant quality, because our focus is the question if additional CO and $NO_2$ instruments are helpful for detecting a $CO_2$ plume. Nonetheless, source attribution is certainly an important step in the algorithm that can be difficult for regions where locations of point sources are not well known. We therefore mention this briefly in the introduction and added the following paragraph to the discussions:

> In our study area, the attribution of detected enhancements to sources was relatively simple, because the locations of the sources were known and plumes were rarely overlapping with plumes from other sources. Source attribution can be more challenging when source location are not precisely known or when several sources are close to each other. For such cases, the algorithm will have to be extended to be operationally applicable.

**Specific comments:**

**Reviewer #2:** P2, P10: Again, even before having a monitoring system, many of cities have no idea of their emissions as they dont have emission inventories.

**Authors reply**: The sentence has been modified as follows: "However, many cities are currently lacking detailed $CO_2$ emission inventories and monitoring systems to evaluate their policies."

**Reviewer #2:** P2, $CO_2$ observation system -> Emission Verification Support System (?)

**Authors reply**: P2, L15 has been changed to "$CO_2$ emission monitoring and verification support system" following the terminology used by Pinty et al. (doi: 10.2760/08644).

**Reviewer #2:** P2, L28: detecting? If you want to stick to detection, Nassar et al. (2017), who sticked to a few cases that worked out (later, the author acknowledged one of the demonstrated power plants was not correct) and Reuter et al. (2019), who used $NO_2$ not just $CO_2$ alone. This sentence is misleading.

**Authors reply**: We slightly modified the paragraph, which reads now: "While some potential for detecting and estimating emissions from $CO_2$ fluxes has been demonstrated for strong $CO_2$ plumes of megacities and large point sources using the Orbiting Carbon Observatory 2 (OCO-2, Crisp et al., 2017) (Nassar et all. 2017; Reuter et al. 2019), it remains a major challenge to accurately determine the location of $CO_2$ plumes, especially of weaker plumes with signal-to-noise ratios near or below the detection limit for single pixels."

**Reviewer #2:** P4, L4: This is not just for weak anthropogenic plumes. For example, Nassar et al. (2017) simply assumed the subset of the OCO2 stripe as a background w/o any investigation. How do you distinguish large and weak CO2 emitters? Is 10Mt CO2/yr the threshold?

**Authors reply**: We removed "weak" from the sentence.

**Reviewer #2:** P5, L8: If its only available in German, I would still request a few sentences summarizing this inventory for many non-German speaking audience including myself. W/o knowing this inventory, we dont get a good sense of what the aut hors discuss at L5-L9.

**Authors reply**:
We added the following sentence to make L5-L9 better understandable:

> "The inventories provide point and area sources separately for different sectors (e.g. industry, heating and road-transport) using Selected Nomenclature for Air Pollution (SNAP) categories."

The Berlin inventory is also described in more detail by Brunner et al. (2019) and in the SMARTCARB final report.

**Reviewer #2**: P10, L28: simplified from?

**Authors reply**: Changed "simplified" to "simple"

**Reviewer #2:** P32, L19: point sources -> coal-fired power plants? Coal-fired power plants are not the only kind for $CO_2$ point sources.

**Authors reply**: Sorry, we are not able to assign the comment to the cited line in the manuscript. Of course, other $CO_2$ point sources exists, although the point sources studied here were all coal-fired power plants.

[revised manuscript text omitted]

- X_BER_BG: concentrations from emissions, fluxes and lateral boundaries excluding emissions from Berlin (= X_TOT – X_BER),

- X_PP_BG: concentrations from emissions, fluxes and lateral boundaries excluding emissions from the six major power plants (= X_TOT – X_PP),

5 where X is $CO_2$, CO or $NO_2$. $NO_2$ concentrations were calculated from $NO_x$ concentrations using an empirical formula used frequently for representing $NO_2$:$NO_x$ ratios downstream of emission sources (Düring et al., 2011). For $NO_2$ only the tracers with a lifetime of 4 hours were used. Note that only the sum of the emissions from the six power plants was simulated but not the power plants individually, which often complicated the analysis due to overlapping plumes. For the analysis, the three-dimensional model fields were vertically integrated to compute column-averaged dry air mole fractions of $CO_2$ (XCO$_2$).

10 Likewise, tropospheric CO and $NO_2$ vertical column densities (VCD) were generated by considering only the model fields below 10 km altitude.

**2.1.2 Satellite instrument scenarios**

The CO2M mission is a proposed constellation of satellites flying in a sun-synchronous low-earth orbit with equator crossing times around 11:30 local time. Each satellite will carry an imaging spectrometer measuring in the near-infrared (NIR) and in

two short-wave infrared spectral bands (SWIR-1 and SWIR-2) for retrieving $CO_2$ as the main payload. The NIR band is used to retrieve information on the dry air column, on surface pressure and on aerosols and clouds. The SWIR-1 and SWIR-2 bands contain weak and strong absorption features of $CO_2$ and provide additional information on aerosols and clouds, especially on thin cirrus clouds. A $CO_2$ retrieval using these three bands is described for example by O'Dell et al. (2012). CO2M is planned
5 to carry also additional instruments for measuring $NO_2$, aerosols and clouds. In an earlier phase, also an instrument measuring CO was considered. The preliminary system concept envisages a pixel size of 4 $km^2$ and swath width of 250 km or more.

For the $CO_2$, CO and $NO_2$ satellite observations, different instrument scenarios were prescribed by ESA for  this study in terms of orbit, spatial resolution and spatial and temporal coverage of the CO2M instrument. In addition, the Sentinel-5 instrument on-board the Meteorological Opera-
10 tional Satellite – Second Generation - A  (MetOp-SG-A) was studied as an alternative platform for CO and $NO_2$ measurements. Sentinel-5  will be an imaging spectrometer measuring, among others, $NO_2$ and CO  columns with a spatial resolution of $7\times7$ $km^2$ and a 2650-km swath. MetOp-SG-A will be also on a sun-synchronous
15  orbit but with different equator crossing times and repeat cycles than the CO2M mission.

[revised manuscript text omitted]